# Quantifying Prediction Consistency Under Fine-Tuning Multiplicity in Tabular LLMs

Faisal Hamman [1]  Pasan Dissanayake [1]  Saumitra Mishra [2]  Freddy Lecue [2]  Sanghamitra Dutta [1]

## Abstract

Fine-tuning LLMs on tabular classification tasks can lead to the phenomenon of *fine-tuning multiplicity* where equally well-performing models make conflicting predictions on the same input. Fine-tuning multiplicity can arise due to variations in the training process, e.g., seed, weight initialization, minor changes to training data, etc., raising concerns about the reliability of Tabular LLMs in high-stakes applications such as finance, hiring, education, healthcare. Our work formalizes this unique challenge of fine-tuning multiplicity in Tabular LLMs and proposes a novel measure to quantify the consistency of individual predictions without expensive model retraining. Our measure quantifies a prediction's consistency by analyzing (sampling) the model's local behavior around that input in the embedding space. Interestingly, we show that sampling in the local neighborhood can be leveraged to provide probabilistic guarantees on prediction consistency under a broad class of fine-tuned models, i.e., inputs with sufficiently high local stability (as defined by our measure) also remain consistent across several fine-tuned models with high probability. We perform experiments on multiple real-world datasets to show that our local stability measure preemptively captures consistency under actual multiplicity across several fine-tuned models, outperforming competing measures.

## 1. Introduction

Large language models (LLMs) are generating significant interest in high-stakes applications, e.g., finance, healthcare, etc., particularly in few-shot classification scenarios. Since many of these sectors rely on tabular data, Tabular LLMs (TabLLMs) is emerging as a research priority (van Breugel & van der Schaar, 2024). Recent studies have shown that TabLLMs perform commendably in scenarios with limited training data due to their transfer learning abilities (Hegselmann et al., 2023; Dinh et al., 2022; Yin et al., 2020; Yan et al., 2024; Wang et al., 2023). However, these models are often fine-tuned from large pre-trained models with millions or billions of parameters on small, proprietary datasets (Hu et al., 2021; Liu et al., 2022). This paucity of training data and large parameter space introduces arbitrariness and inconsistency across fine-tuned model variants, raising concerns about their trustworthy adoption in high-stakes applications.

One imminent challenge for TabLLMs is *fine-tuning multiplicity* where multiple well-performing models fine-tuned from the same pre-trained LLM under slightly varying conditions (e.g., different seed, hyperparameters, or minor changes in training data) produce conflicting predictions for the same inputs. This concept is closely related to predictive multiplicity, often referred to as the Rashomon effect in the context of decision trees and neural networks (Marx et al., 2020; Breiman, 2003; Hsu & Calmon, 2022). While multiplicity has also been observed in LLMs for text classification (Gomez et al., 2024), we are particularly interested in fine-tuning multiplicity in TabLLMs (essentially minor model variations) due to their relevance in high-stakes classification tasks. E.g., in areas like finance (Yin et al., 2023) and healthcare (Wang et al., 2024b; Chen et al., 2023b; Kim et al., 2024), arbitrary and conflicting predictions on the same input under minor model variations can lead to confusion, reputational risk, and distrust among stakeholders.

Aside from the inherent need for predictions to be consistent to minor model variations (due to seed or hyperparameters), TabLLMs deployed by institutions may also need to be updated for various reasons, e.g., to retrain on a few additional data points (Wu et al., 2024), or even remove few data points for privacy. Regulatory frameworks like the GDPR (Voigt, 2017) introduce the *right to be forgotten* which allows unlearning an individual's data upon request, potentially leading to model updates. These model updates could, in turn, impact previously issued predictions. Fine-tuning multiplicity also paves the way for fairwashing and explanation bias (Black et al., 2022; Sokol et al., 2023; Rudin et al.,

---

[1] Department of Electrical and Computer Engineering, University of Maryland, College Park [2] JPMorgan Chase AI Research. Correspondence to: Faisal Hamman <fhamman@umd.edu>.

*Proceedings of the $42^{nd}$ International Conference on Machine Learning*, Vancouver, Canada. PMLR 267, 2025. Copyright 2025 by the author(s).

2024), making quantifying consistency under fine-tuning multiplicity an important and practically relevant problem.

Existing approaches to measure multiplicity in machine learning often involve retraining and ensembling multiple models (Marx et al., 2020). However, such approaches can be computationally expensive for LLMs due to large parameter sizes. This raises a key question: *Can we **preemptively** quantify the consistency of individual predictions under fine-tuning multiplicity without actual retraining and ensembling?* To address this question, we propose a novel measure, termed *local stability*, which leverages the model's local behavior around each input data point in the embedding space to estimate the prediction's susceptibility to multiplicity. Our contributions are summarized as follows:

- **Study the intriguing nature of fine-tuning multiplicity in TabLLMs.** We first demonstrate that prediction inconsistency exists when we actually fine-tune several models from the same pre-trained model, as observed through existing multiplicity measures such as *Arbitrariness*, *Discrepancy*, *Pairwise Disagreement*, as well as two of our proposed multiplicity measures, *Prediction Variance*, and *Range* (defined in Section 2). We also visualize the decision boundary for several TabLLMs fine-tuned for a simple classification task and unravel an interesting "noise" pattern: unlike neural network classifiers which typically have locally-smooth decision boundaries, Tabular LLMs show abrupt and impulsive variations (see Fig. 2). Thus, a model having high confidence in a prediction alone does not guarantee its consistency under fine-tuning multiplicity.

- **A measure to quantify prediction consistency under fine-tuning multiplicity.** We introduce a novel measure, termed *local stability* (see Definition 5), to quantify the consistency of model predictions under fine-tuning multiplicity without retraining several models. Given an input $\mathbf{x}$ and a model's prediction probability for a class $c$, i.e., $f_c(\mathbf{x}) \in [0, 1]$, our measure is $S_{k,\sigma}(\mathbf{x}, f_c) = \frac{1}{k} \sum_{\mathbf{x}_i \in N_{\mathbf{x},k}} (f_c(\mathbf{x}_i) - |f_c(\mathbf{x}) - f_c(\mathbf{x}_i)|)$, where $N_{\mathbf{x},k}$ is a set of $k$ points sampled independently from a distribution over a hypersphere of radius $\sigma$ centered at $\mathbf{x}$. This measure uses the input's local neighborhood (in the embedding space) to inform the local stability, capturing both the mean model confidence in the neighborhood and the variability in confidence in that region.

- **Probabilistic guarantees on consistency over a broad class of fine-tuned models.** We provide a theoretical guarantee (see Theorem 1) that predictions with sufficiently high local stability (as defined by our measure) will remain consistent with high probability over a *broad range of equally-well-performing fine-tuned models*. To derive this guarantee, we make some mild assumptions on the behavior of this fine-tuned model class (see Assumption 1). Our proof leverages Hoeffding's Inequality (see Lemma 2).

- **Experimental results.** We show that our Local Stabil-

ity measure (computed preemptively without retraining or ensembling) is quite well-aligned with the consistency of data points under actual fine-tuned multiplicity for several datasets, namely, the `German Credit`, `Bank`, `Heart`, `Car`, `Diabetes`, and `Adult` datasets (Kahn; Hofmann, 1994; Becker & Kohavi, 1996). We employ the BIG-SCIENCE T0 encoder-decoder model (Sanh et al., 2021) and Google FLAN-T5 (Chung et al., 2024), fine-tuned via the T-Few recipe (Liu et al., 2022), and LoRA (Hu et al., 2021). For each case, we empirically evaluate the extent of fine-tuning multiplicity, and also study how our local stability measure $S_{k,\sigma}(\mathbf{x}, f)$, (measured only using one model $f$) can preemptively capture consistency under fine-tuning multiplicity better than competing measures including prediction confidence alone.

***Related Works:*** LLMs for tabular data is a growing area of research (Yin et al., 2020; Li et al., 2020; Narayan et al., 2022; Borisov et al., 2022; Bertsimas et al., 2022; Onishi et al., 2023; Zhang et al., 2023; Wang et al., 2023; Sui et al., 2024; Yan et al., 2024; Yang et al., 2024). While neural networks and gradient-boosted trees perform well with tabular data when ample labeled data is available, their effectiveness drops considerably in data-scarce scenarios. In contrast, LLMs can leverage their *reasoning*, in-context learning, and pre-trained knowledge to maintain strong performance even on tiny tabular datasets (Hegselmann et al., 2023). Dinh et al. (2022) proposes LIFT, a method for adapting LLMs to non-language classification and regression tasks without changing the model architecture or loss function. Hegselmann et al. (2023) studies the use of LLMs for zero-shot and few-shot classification of tabular data and finds that this method outperforms previous deep-learning-based approaches and is competitive with traditional baselines like gradient-boosted trees. Wang et al. (2024b) presents MediTab, a method that uses LLMs to combine different medical datasets, significantly improving predictions for patient and trial outcomes. Tabular LLMs have also been applied in other high-stakes domains (Chen et al., 2023b; Kim et al., 2024; Li et al., 2023; Yin et al., 2023). Yin et al. (2023) presents FinPT, an LLM based approach to financial risk prediction. We refer to Fang et al. (2024) for a more detailed survey on LLMs for tabular data.

Breiman (2003) introduced the idea that models can differ significantly while achieving similar average performance, known as the Rashomon effect. Marx et al. (2020) highlighted the prevalence of arbitrary decisions in simple classification problems, calling this predictive multiplicity. Creel & Hellman (2022) discuss the harms of predictive multiplicity and arbitrary decisions. Methods such as Tree-Farms (Xin et al., 2022), CorelsEnum (Mata et al., 2022), and RashomonGB (Hsu et al.) provide tools to enumerate models in the Rashomon set for different hypothesis spaces. Efforts to leverage model multiplicity beneficially

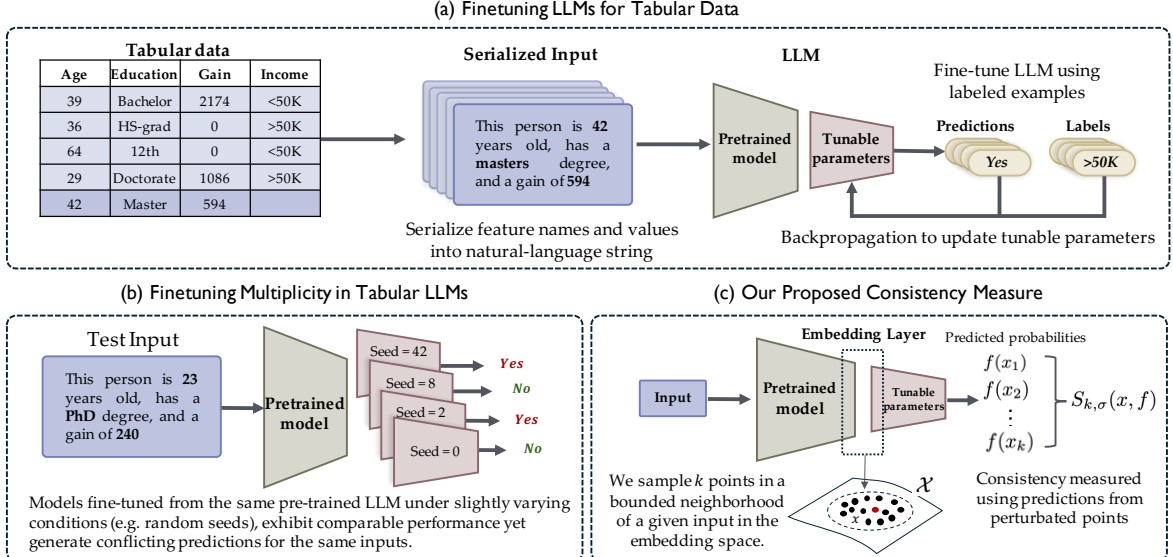

*Figure 1.* (*a*) illustrates the process of fine-tuning LLMs for Tabular data using few labeled examples (Hegselmann et al., 2023; Dinh et al., 2022). (*b*) demonstrates the concept of finetuning multiplicity. Models fine-tuned from the same pre-trained LLM under slightly varying conditions, such as different random seeds, can exhibit comparable performance metrics but may yield conflicting predictions for the same input. (*c*) introduces our proposed local stability measure designed to quantify the consistency of individual predictions without requiring the retraining of multiple models. By sampling points in a bounded neighborhood around a given input in the embedding space, the consistency measure $S_{k,\sigma}(\mathbf{x}, f)$ informs a prediction's susceptibility to multiplicity.

while addressing its implications have been explored by (Black et al., 2022; Fisher et al., 2019; Xin et al., 2022; Coston et al., 2021). Model multiplicity in fairness and explainability are examined by Sokol et al. (2022); Hamman et al. (2023; 2024); Black et al. (2021); Dutta et al. (2022); Pawelczyk et al. (2020). Watson-Daniels et al. (2023); Hsu & Calmon (2022) offered a framework for measuring predictive multiplicity in machine learning models, however, this involves retraining several models, with the exception of (Hsu et al., 2024) who propose a drop-out based approach to explore the Rashomon set for neural networks. Model multiplicity under fine-tuning has not been extensively studied in TabLLMs. The closest work is Gomez et al. (2024), which empirically investigates prediction arbitrariness for text classification (online content moderation). In this work, we isolate and examine a specific form of multiplicity in TabLLMs that focuses on minor model variations due to fine-tuning from the same pre-trained LLM (see Section 2). We leverage the embedding space of LLMs to preemptively quantify consistency under fine-tuning multiplicity without expensive retraining (see Section 3). There are also other alternate directions in robustness literature that have focused on aspects other than multiplicity such as out-of-distribution generalization, adversarial examples, and uncertainty estimation (Djolonga et al., 2020; Han et al., 2023).

**Preliminaries:** We consider a classification task for a tabular dataset $D = \{(\mathbf{x}_i, y_i)\}_{i=1}^n$, where each $\mathbf{x}_i$ is a $d$-dimensional feature vector (rows of a tabular input), and each label $y_i$ is binary, $y_i \in \{0, 1\}$. We focus on $n$-shot classification where a pre-trained model is fine-tuned on a limited number of training examples $n$.

*Serialization of Tabular Data for LLMs:* To apply LLMs to tabular data, it is crucial to transform the data into a natural text format. This process, known as serialization, involves converting the table rows into a text string that includes both the column names and their corresponding values (Yin et al., 2020; Jaitly et al., 2023; Hegselmann et al., 2023; Dinh et al., 2022). The resultant serialized string is combined with a task-specific prompt to form the input for the LLM. There have been various proposed methods for serialization, and this is still a topic of active research (Jaitly et al., 2023). Among the serializations we have examined are: list template (a list of column names and feature values), and text template ( The <column name> is <value> ). The training process uses the natural-language outputs of the LLM, mapped to valid classes in the target space, as part of fine-tuning (see Fig. 1). To clarify, table values are serialized into serialize($\mathbf{x}$) and then transformed into a format understandable by the LLM, tokenize(serialize($\mathbf{x}$)), which is an embedding. Since these transformations are one-to-one mappings, we denote the embedded form of input $\mathbf{x}$ also as $\mathbf{x} \in \mathcal{X}$ to represent $\mathbf{x}$ in the embedding space. This allows us to simplify the notation and directly use $\mathbf{x}$ to refer to the input values in the embedding space.

## 2. Multiplicity in Fine-Tuned Tabular LLMs

Let $f(\cdot) = [f_1(\cdot), f_2(\cdot), \ldots, f_C(\cdot)] : \mathcal{X} \to \Delta^C$ denote an LLM that performs multi-class classification over $C$ classes, where $\Delta^C$ is the $C$-dimensional probability simplex (e.g., softmax outputs). Let $\mathcal{F}$ denote a broad class of equally-well-performing fine-tuned models (a set of competing fine-tuned models as measured by the accuracy), i.e, $\mathcal{F}_\delta = \{f : err(f) \leq err(f_0) + \delta\}$ where $err(f_0) = \frac{1}{N} \sum_{i=1}^{N} \mathbb{I}[\hat{f}_0(\mathbf{x}_i) \neq y_i]$ for a reference model $f_0$ (with satisfactory accuracy) and test dataset with $N$ examples. Here, $\hat{f}(\mathbf{x}) = \arg\max_{c \in [C]} f_c(\mathbf{x})$ denotes the predicted label. This is a set of fine-tuned models that perform just as well as the reference baseline classifier $f_0$, where $\delta \in (0, 1)$ is the error tolerance. The appropriate choice of $\delta$ is application-dependent (Marx et al., 2020).

**Fine-tuning Multiplicity.** We study the nature of multiplicity that arises in LLMs when fine-tuned for tabular tasks. To illustrate fine-tuning multiplicity, we conduct experiments using synthetic 2D data (see Fig. 2). While fine-tuning an LLM on such data might seem excessive, it provides a clear visualization of the phenomenon. We fine-tune several competing models using the text template ( The <column name> is <value> ) and varying only the random training seed. We reveal that fine-tuning LLMs on such non-language tasks exhibit noisy and non-smooth decision boundaries, even in regions where the model is expected to confidently predict a specific class. We hypothesize that this noisy behavior is likely because LLMs are optimized for capturing complex language structures. When fine-tuned on tabular data tasks, which often involve both text and numeric values, LLMs leverage their pre-trained knowledge but still exhibits instabilities.

**Evaluating Fine-tuning Multiplicity.** To evaluate the extent of multiplicity on actual fine-tuned models, we now introduce specific empirical metrics that assess how predictions may vary across different fine-tuned models.

**Definition 1** (Arbitrariness (Gomez et al., 2024)). *Arbitrariness over set $\mathcal{F}_\delta$ measures the extent of conflicting predictions across the model space for a given set of inputs $\{\mathbf{x}_1, \ldots, \mathbf{x}_n\}$. It is defined as: $A_\delta = \frac{1}{n} \sum_{i=1}^{n} \mathbb{I}[\exists f, f' \in \mathcal{F}_\delta, : \hat{f}(\mathbf{x}_i) \neq \hat{f}'(\mathbf{x}_i)]$.*

Arbitrariness generalizes the *Ambiguity* measure which computes the fraction of points where at least one model in $\mathcal{F}_\delta$ disagrees with a reference model (Marx et al., 2020). Abitrariness measures the percentage of points that receive conflicting predictions from any two models within the set $\mathcal{F}_\delta$. Arbitrariness can also be defined on a single input, i.e., $A(\mathbf{x}_i) = \mathbb{I}[\exists f, f' \in \mathcal{F}_\delta, : \hat{f}(\mathbf{x}_i) \neq \hat{f}'(\mathbf{x}_i)]$.

**Definition 2** (Discrepancy). *Discrepancy quantifies the maximum proportion of conflicting predictions between a reference model and any competing model in the set:*

$$D_\delta(f_0) := \max_{f \in \mathcal{F}_\delta}(\frac{1}{n} \sum_{i=1}^{n} \mathbb{I}[\hat{f}(\mathbf{x}_i) \neq \hat{f}_0(\mathbf{x}_i)]).$$

Discrepancy measures the maximum number of predictions that could change if a reference model is replaced with a competing model. This means that, in practice, altering multiple predictions requires that all conflicting predictions come from a single competing model.

**Definition 3** (Pairwise Disagreement (Black et al., 2022)). *Pairwise Disagreement assesses the variability among models by measuring the proportion of instances where pairs of models within the competing set disagree: $PD_\delta(\mathbf{x}) := \frac{1}{|\mathcal{F}_\delta|(|\mathcal{F}_\delta|-1)} \sum_{f^i, f^j \in \mathcal{F}_\delta, f^i \neq f^j} \mathbb{I}[\hat{f}^i(\mathbf{x}) \neq \hat{f}^j(\mathbf{x})]$.*

Since existing measures of multiplicity focus on predicted labels, we propose two more nuanced measures that leverage the predicted probabilities of model outputs:

**Definition 4** (Prediction Variance). *PV measures the variability of the model outputs for a given input $\mathbf{x}$ and class $c$ across different models in the set $\mathcal{F}_\delta$: $PV_\delta(\mathbf{x}) := \frac{1}{|\mathcal{F}_\delta|} \sum_{f \in \mathcal{F}_\delta} (f_c(\mathbf{x}) - \frac{1}{|\mathcal{F}_\delta|} \sum_{f' \in \mathcal{F}_\delta} f'_c(\mathbf{x}))^2$.*

Unlike threshold-based measures, Prediction Variance captures variability in predicted probabilities. We also define *Prediction Range* to quantify the maximum spread in predicted probabilities: $PR_\delta(\mathbf{x}) := \max_{f \in \mathcal{F}_\delta} f_c(\mathbf{x}) - \min_{f \in \mathcal{F}_\delta} f_c(\mathbf{x})$ (also see Watson-Daniels et al. (2023)).

For brevity, from now on we use $f(\mathbf{x})$ to refer to the predicted probability for the specific class of interest (typically the predicted class for $\mathbf{x}$), rather than the full vector of probabilities. Thus, $f(\mathbf{x}) \in [0, 1]$ will be a scalar.

## 3. A novel measure to preemptively capture prediction consistency

Our objective is to define a measure, denoted as $S(\mathbf{x}, f)$, for an input $\mathbf{x}$ and a given fine-tuned model $f$, that would preemptively quantify the consistency of the prediction $\hat{f}(\mathbf{x})$ over a broad class of equally-well-performing fine-tuned models. We desire that the measure $S(\mathbf{x}, f)$ should be high if the predictions for the input $\mathbf{x}$ is consistent across this broad class of fine-tuned models (see Fig. 1).

**Candidate Measure: Prediction confidence** $S(\mathbf{x}, f) := f(\mathbf{x})$. While the prediction probability of a model $f(\cdot)$ offers insights into its confidence in predicting a given class, they are insufficient for assessing consistency under fine-tuning multiplicity (see Table 2, Fig. 3, i.e., data point with high $f(\mathbf{x})$ or confidence can still be susceptible to multiplicity). In our synthetic data experiments (see Fig. 2), we also observe that noisy behaviors emerge in regions where the model should be confident in its predictions, leading to conflicting outcomes across various fine-tuned models. This indicates that relying solely on an input $\mathbf{x}$ may not provide a reliable assessment of consistency. To address this, we

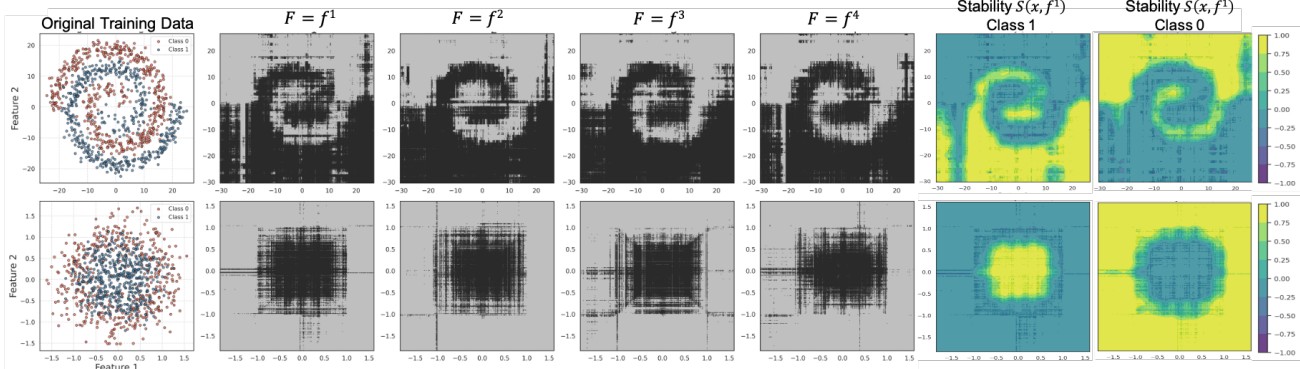

*Figure 2.* **Decision boundaries for multiple fine-tuned models of an LLM on synthetic datasets**. We fine-tuned several models by only changing the random training seed. All models achieve comparable training loss and accuracy, yet they converge to different functions, exhibiting intriguing noisy patterns (a phenomenon absent in models like neural networks which are typically locally-smooth). Interestingly, these noisy behaviors appear even in regions where the model is expected to confidently predict a specifc class. Observe the location and shape of these noisy patterns vary unpredictably across the various fine-tuned models, making them a possible factor contributing to prediction multiplicity. This highlights that model predictions alone may be unreliable and motivates our perturbation-based approach to quantify multiplicity. The last two plots illustrate the local stability measure applied to model $f^1$ across classes 0 and 1, i.e., $S(\cdot, f^1)$. The local stability measure effectively highlights regions where predictions are reliable (indicated by bright yellow color) and areas where predictions may be unstable.

propose a perturbation-based approach that leverages the local neighborhood around the input $\mathbf{x}$ in the embedding space, ultimately leading to our measure of local stability.

### 3.1. Proposed Measure: Local Stability

**Definition 5** (Local Stability). *For a given data point $\mathbf{x}$, let $f(\mathbf{x})$ represent the predicted probability (e.g., softmax logits) from a model $f$. The local stability is defined as:*

$$S_{k,\sigma}(\mathbf{x}, f) = \frac{1}{k} \sum_{\mathbf{x}_i \in N_{\mathbf{x},k}} f(\mathbf{x}_i) - \frac{1}{k} \sum_{\mathbf{x}_i \in N_{\mathbf{x},k}} |f(\mathbf{x}) - f(\mathbf{x}_i)|,$$

$N_{\mathbf{x},k} = \{\mathbf{x}_1, \mathbf{x}_2, \ldots, \mathbf{x}_k\} \subset B(\mathbf{x}, \sigma) = \{\mathbf{x}' \in \mathcal{X} \mid \|\mathbf{x}' - \mathbf{x}\|_2 < \sigma\}$ *is a set of $k$ points sampled independently from a distribution over a hypersphere of radius $\sigma$ centered at $\mathbf{x}$.*

Our local stability measure is tied to the confidence in predicting a specific class (i.e., the probability values derived from softmax logits), and not on the predicted labels. It quantifies the stability of a model's predictions using the local neighborhood around an input $\mathbf{x}$. The first term, $\frac{1}{k} \sum_{\mathbf{x}_i} f(\mathbf{x}_i)$, captures the average confidence of the model in this region. The second term, $\frac{1}{k} \sum_{\mathbf{x}_i} |f(\mathbf{x}) - f(\mathbf{x}_i)|$, penalizes variability by measuring how much the predictions fluctuate within the neighborhood. By subtracting this variability from the local average confidence, the measure ensures that high scores are assigned to predictions with high local neighborhood prediction confidence and low variability. This formulation is motivated by our observations on synthetic data, where models exhibited irregular, non-smooth decision boundaries despite high confidence in

certain regions (see Fig. 2). See App. A for more intuitions and properties of stability measure.

### 3.2. Theoretical Guarantees on Consistency

We present theoretical insights that motivate our proposed stability measure $S_{k,\sigma}(\mathbf{x}, f)$, in quantifying the consistency of predictions across a broad class of fine-tuned models.

Let $\bar{f}(\cdot; \bar{W})$ represent a target function and $f_0(\cdot; W)$ denote a pre-trained model. Parameter-efficient fine-tuning methods such as Low-Rank Adaptation (LoRA) adds a low-rank updates into the weight matrices of a frozen pre-trained model to effectively approximate the target function (Hu et al., 2021). In this framework, the fine-tuned model is represented as $F(\cdot; W + \Delta W)$, where the low-rank weight updates $\Delta W$ are added to the frozen model. We let the fine-tuned model be a random variable, $F(\cdot; W + \Delta W)$, where the randomness arises from the distribution of low-rank weights $\Delta W \in \mathcal{W}$. For clarity, we use capital letters (e.g., $F, X_i, Z$) to denote random variables, while lower-case letters (e.g., $\mathbf{x}_i, f, \epsilon$) indicate specific realizations. For brevity, we omit the weight parametrization.

**Assumption 1.** *Let $\bar{f}$ and $F$ denote the target and adapted models, respectively. We assume:*
- $\mathbb{E}[F(X)|X = \mathbf{x}] = \mathbb{E}[F(\mathbf{x})] = \bar{f}(\mathbf{x})$, *i.e., $F(\mathbf{x})$ is an unbiased estimator of $\bar{f}(\mathbf{x})$.*
- $\mathbb{E}_X[\|F(X) - \bar{f}(X)\| \mid F = f] \leq \alpha$, *and the expected norm of its gradient $\mathbb{E}_X[\|\nabla(F - \bar{f})(X)\| \mid F = f] \leq t$, where $X$ is drawn independently from a distribution over the hypersphere $B(\mathbf{x}, \sigma)$.*

• *F and $\bar{f}$ are twice differentiable with Hessians bounded by L, i.e., $\|\nabla^2 F(\mathbf{x})\|, \|\nabla^2 \bar{f}(\mathbf{x})\| \leq L$.*

Our Assumptions are motivated by recent work by (Zeng & Lee, 2023) which provides key theoretical insights into the expressive power of LoRA adaptation for transformer models. Specifically, we assume that the fine-tuned model $F(\mathbf{x})$ is an unbiased estimator of the target function $\bar{f}(\mathbf{x})$, meaning that across fine-tuned variations, the expectation of the model remains centered around the true function. Their results also establish the expected error over a bounded region, given by $\mathbb{E}_X[\|F(X) - \bar{f}(X)\| | F = f] \leq \alpha$. Under certain conditions, they show the existence of adapter weights $\Delta W$ such that $\alpha \to 0$. Additionally, we assume that the expected gradient norm is bounded by $t$, implying that while gradients may vary across fine-tuned models, they remain close to the target function in expectation over a local region.

**Theorem 1** (Probabilistic Guarantee). *Given a data point $\mathbf{x}$, a target model $\bar{f}$ and local stability measure $S_{k,\sigma}(\mathbf{x}, F)$. Under Assumption 1, and for all $\epsilon > \epsilon'$, where $\epsilon' = 2(\alpha + t\sigma) + \mathcal{O}(L\sigma^2)$. We have:*

$$\Pr\left(\bar{f}(\mathbf{x}) \geq S_{k,\sigma}(\mathbf{x}, F) - \epsilon\right) \geq 1 - \exp\left(\frac{-k\epsilon^2}{32}\right) \quad (1)$$

**Theoretical Guarantee Interpretation.** Our stability measure $S(\mathbf{x}, f)$ provides a probabilistic guarantee that if a data point $\mathbf{x}$ has a sufficiently high local stability score on a random model $F$, sampled from a broad class of equally-well-performing fine-tuned models, then the prediction on the target model $\bar{f}(\mathbf{x})$ will be at least $S(\mathbf{x}, f) - \epsilon$ with high probability. For example, if $S(\mathbf{x}, f) = 0.8$, we can be confident that $\bar{f}(\mathbf{x})$ will be at least $0.8 - \epsilon$ with *high* probability (i.e, the prediction will remain on the positive predicted side). This implies that high local stability scores are indicative of consistent predictions. The probability of the bound holding increases exponentially with the sample size $k$. Conversely, a low stability score does not provide significant information about the prediction's behavior, as it does not guarantee a lower bound on the prediction.

**Goodness of Model Class.** The term $\epsilon'$ is indicative of the quality or goodness of the fine-tuned model class. A small $\epsilon'$ indicates a well-behaved model class, suggesting that different fine-tuned models produce similar outputs in expectation within the local neighborhood of $\mathbf{x}$ even if predictions might vary for a given data point. Similar behavior is visualized in Figure 2, where, despite the presence of noisy variations in the decision boundaries, the local predictions around a given point remain relatively consistent across models. This behavior is expected since these models are derived from the same pre-trained model and trained with the goal of achieving similar accuracy on the dataset. In this case, *our local stability measure provides an informative lower bound on the predictions $\bar{f}(\mathbf{x})$ with a certifiably small gap.*

Conversely, a large $\epsilon'$ indicates a more erratic model class. In this case, our bound becomes less informative, and the local stability measure might perform poorly for a given point. We interpret our results as follows: The model class is not well-behaved; thus, one cannot certify a small gap between $\bar{f}(\mathbf{x})$ and our proposed measure. We do not provide guarantees for all types of model changes, as this would be challenging with only a single model. For example, if fine-tuned models do not achieve sufficient accuracy, encounter significant variations in hyperparameter choices, or large changes in the training data, $\epsilon'$ is likely to be large. Our focus is on multiplicity that arises due to randomness in training, such as changes in the training seed or minor adjustments in training settings (i.e., a broad class of equally-well-performing fine-tuned models). In our evaluations, we do not assume any specific values for $\epsilon'$ and consider regular fine-tuned models without imposing any theoretical constraint. The proof of Theorem 1 is provided in App. B.

## 4. Empirical Results

In this section, we experiment across different datasets to *(i)* quantify the prevalence of fine-tuning multiplicity in Tabular LLMs, and *(ii)* validate the effectiveness of our proposed measure in quantifying the consistency of predictions over a broad range of equally-well-performing fine-tuned models.

**Datasets and Serialization.** Our experiments utilize the `Diabetes` (Kahn), `German Credit` (Hofmann, 1994), `Bank` (Moro et al., 2014), `Heart`, `Car`, and `Adult` datasets (Becker & Kohavi, 1996), serialized using the Text Template, i.e., tabular entry is converted into a natural language: `The <column name> is <value>`. This approach helps align the inputs with the training distribution of LLMs, enhancing their performance in few-shot scenarios (Hegselmann et al., 2023; Dinh et al., 2022).

**Models and Fine-tuning Methods.** We use the BIG-SCIENCE T0 (Sanh et al., 2021) and Google FLAN-T5 (Chung et al., 2024) encoder-decoder models as our pretrained LLMs. T0 is specifically pre-trained for zero-shot generalization through multitask learning. FLAN-T5 is instruction fine-tuned on a diverse range of tasks, achieving strong performance in few-shot settings. These make both models well-suited for our experiments. For fine-tuning, we adopt the T-Few recipe (Liu et al., 2022), known for its effectiveness in few-shot learning, and LoRA (Hu et al., 2021). Detailed setup can be found in App. C.3.

**Evaluating Extent of Fine-tuning Multiplicity.** We measure the extent of fine-tuning multiplicity across the various datasets and fine-tuning methods, we use the multiplicity evaluation metrics (see Section 2). To evaluate these multiplicity metrics across our datasets, we fine-tune 40 models on Tfew recipe and LoRA using different random seeds and

*Table 1.* Evaluated Multiplicity for Different Datasets and Number of Shots on BIGSCIENCE T0. Evaluated on 40 fine-tuned models on T-Few recipe using different random seeds. Multiplicity observed in predictions across different fine-tuned model, even when models exhibit similar accuracy (in this setting $\delta = 0.02$). Fine-tuning using LoRA achieves results in the same ballpark (see LoRA Table 6 in App. C)

| Dataset | No. | Multiplicity Evaluation Metrics (BIGSCIENCE T0) | | | | | |
|---|---|---|---|---|---|---|---|
| | Shots | Arbit. | Disc. | Avg. Pair. Disag. | Avg. Pred. Variance | Avg. Pred. Range | Avg. Model Accuracy |
| Adult | 64 | 10% | 9% | 7% | 0.01 | 0.10 | 83% |
| | 128 | 10% | 7% | 8% | 0.01 | 0.10 | 84% |
| | 512 | 11% | 8% | 7% | 0.01 | 0.12 | 85% |
| German | 64 | 18% | 10% | 6% | 0.01 | 0.20 | 71% |
| | 128 | 17% | 11% | 6% | 0.01 | 0.16 | 71% |
| | 512 | 23% | 12% | 7% | 0.02 | 0.23 | 72% |
| Diabetes | 64 | 29% | 18% | 10% | 0.04 | 0.31 | 71% |
| | 128 | 13% | 17% | 11% | 0.03 | 0.13 | 72% |
| | 512 | 16% | 16% | 10% | 0.02 | 0.18 | 78% |
| Bank | 64 | 11% | 9% | 6% | 0.01 | 0.31 | 66% |
| | 128 | 15% | 8% | 7% | 0.03 | 0.22 | 75% |
| | 512 | 14% | 8% | 7% | 0.02 | 0.16 | 81% |
| Heart | 64 | 6% | 4% | 2% | 0.01 | 0.05 | 78% |
| | 128 | 9% | 4% | 3% | 0.01 | 0.10 | 83% |
| | 512 | 18% | 7% | 5% | 0.01 | 0.19 | 82% |
| Car | 64 | 19% | 10% | 6% | 0.01 | 0.18 | 81% |
| | 128 | 16% | 7% | 5% | 0.01 | 0.14 | 86% |
| | 512 | 8% | 4% | 2% | 0.01 | 0.09 | 94% |

*Table 2.* This table reports the Absolute Spearman Correlation between the stability measure and various multiplicity evaluation metrics for 128 shots on the datasets. In most cases, our stability measure $S_{k,\sigma}(x,f)$ shows a higher correlation with these multiplicity measures compared to predicted probabilities and drop-out method, indicating that the stability measure $S_{k,\sigma}(x,f)$ better informs about the multiplicity than other measures. See full Table 8 with 64 and 512 shot cases in App. C.

| Dataset | Number of Shots | Measure | Arbit. | Pairwise Disag. | Prediction Variance | Prediction Range |
|---|---|---|---|---|---|---|
| Adult | 128 | Pred. Prob. | 0.67 | 0.62 | 0.30 | 0.54 |
| | | Drop-Out | 0.74 | 0.83 | 0.69 | 0.81 |
| | | Stability | **0.80** | **0.96** | **0.84** | **0.91** |
| German | 128 | Pred. Prob. | **0.57** | **0.57** | 0.86 | 0.86 |
| | | Drop-Out | 0.50 | 0.56 | 0.74 | 0.84 |
| | | Stability | 0.54 | 0.54 | **0.87** | **0.87** |
| Diabetes | 128 | Pred. Prob. | 0.88 | 0.93 | **0.93** | **0.95** |
| | | Drop-Out | 0.89 | 0.92 | 0.92 | 0.94 |
| | | Stability | **0.92** | **0.95** | 0.93 | **0.95** |
| Bank | 128 | Pred. Prob. | 0.54 | 0.57 | 0.73 | 0.62 |
| | | Drop-Out | 0.62 | 0.70 | 0.75 | 0.51 |
| | | Stability | **0.79** | **0.84** | **0.87** | **0.86** |
| Heart | 128 | Pred. Prob. | 0.61 | 0.46 | 0.50 | 0.26 |
| | | Drop-Out | 0.64 | 0.76 | 0.74 | 0.83 |
| | | Stability | **0.89** | **0.90** | **0.97** | **0.87** |
| Car | 128 | Pred. Prob. | 0.56 | 0.26 | 0.29 | 0.01 |
| | | Drop-Out | 0.63 | 0.66 | 0.57 | 0.52 |
| | | Stability | **0.97** | **0.91** | **0.93** | **0.94** |

test on a sample set. Here are the experiments we conducted:

• We evaluate multiplicity on the BIGSCIENCE T0 model fine-tuned using *T-Few* (see Table 1).
• We evaluate multiplicity on BIGSCIENCE T0 fine-tuned using *LoRA* (see Table 6 in App. C).
• We evaluate multiplicity on FLAN-T5 model fine-tuned using *T-Few* (see Table 7 in App. C).

**Comparing Local Stability Measure to Evaluated Multiplicity.** We assess the utility of our proposed stability measure $S_{k,\sigma}(\mathbf{x}, f)$ in informing the presence of fine-tuning multiplicity. This utility is measured using the Spearman correlation coefficient (see Definition 6), between our stability $S_{k,\sigma}(\mathbf{x}, f)$ (estimated on just one model) and the evaluated multiplicity (evaluated on several finetuned models), e.g., Spearman$(S_{k,\sigma}(\mathbf{x}, f), PV_\delta(\mathbf{x}))$ across the test set. Here, our local stability measure is taken with respect to the model's predicted class for $\hat{f}(\mathbf{x})$.

*Baselines:* For comparison, we include the following baselines: *1) Prediction probability* $f(\mathbf{x})$ which measures the confidence of the model in predicting a given class. *2) Binary Drop-Out Method* (Hsu et al., 2024): Since there are no other baselines, we adapt this Drop-Out method for TabLLMs. This method drops random weights of the model to explore models in the Rashomon set (i.e., set of competing models) without retraining several models. For a fair comparison, we compare our method (sampling $k$ points in the neighborhood of our data point in the embedding space, and computing the local stability measure) to theirs (averag-

ing the predictions of $k$ models with different dropped-out weights). Note that these require the same number of inferences, hence complexity for both methods are around the same. *Here are the experiments we conducted*:

• We plot the evaluated multiplicity against our stability measure, predicted probabilities, and the drop-out method. See Figure 3 for illustration on the Adult 128 shot (BIGSCIENCE T0 model). For other dataset refer to Figure 6, 7, 8 in App. C.

• We compute the absolute spearman correlation between the stability measures and various multiplicity evaluation metrics (128-shot setting on all datasets presented in Table 2). Results on BIGSCIENCE T0 model with 64 and 512 shots are presented in Table 8 in App. C. Results for FLAN-T5 model are presented in Table 9 in App. C.

**Ablations and Hyperparameter Selection.** Theorem 1 indicate that increasing the sample size $k$ exponentially improves the probability that the stability guarantee holds. However, this also increases the computational cost of model inference. We use $k = 30$, the maximum number that fits into one inference pass on the GPU.

For the *neighborhood radius* $\sigma$, we sampled perturbed points from a truncated Gaussian distribution with a variance of 0.01, which consistently performed well across all experiments. To guide the choice of $\sigma$, we suggest the following data-driven approach. (1) Compute Pairwise Distances: For all training samples, calculate the median distance $d_{med}$ between each point and its k-nearest neighbors (e.g. $k = 5$)

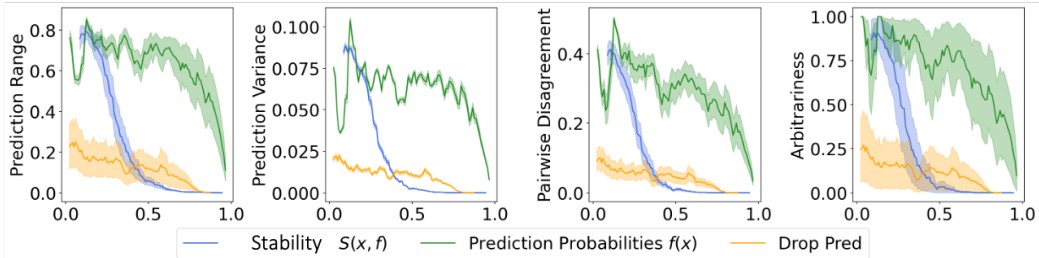

*Figure 3.* Evaluated multiplicity (assessed on 40 retrained models) versus our stability measure, predicted probabilities, and drop-out method (evaluated on one model) for the 128-shot setting on the `Adult` dataset. The plots demonstrate that high local stability values correspond to low multiplicity across various multiplicity evaluation metrics. Also, observe that high predicted probability values (i.e., high prediction confidence) do not imply low multiplicity. Our stability measure provides better insight into the multiplicity of predictions compared to the predicted probabilities or drop-out prediction. App. C for visualizations on other datasets.

in the embedding space. (2) Set $\sigma$ as a fraction of $d_{med}$ (e.g., $\sigma = 0.1 d_{med}$). This captures the natural scale of the data while ensuring perturbations stay within the local neighborhood. This mirrors neighborhood-scale hyperparameters used in clustering, kernel methods (Ester et al., 1996; Cortes & Vapnik, 1995), and certified robustness (Cohen et al., 2019; Salman et al., 2019), which similarly rely on training data pairwise distances to set their parameters.

We used error tolerance $\delta = 0.02$, corresponding to a $2\%$ margin of accuracy deviation. Evaluating multiplicity by refining multiple models is computationally expensive. Thus, we limited our study to 40 models. For the drop-out rate in the baseline, we use $p = 0.1$ following the recommendation in Hsu et al. (2024). To evaluate the impact of varying key parameters, *we conducted the following ablation studies*:

• We perform an ablation study on the sample size $k$, observing improved performance with increasing $k$. Detailed results are provided in Table 11 in App. C.

• We explore the effect of varying the neighborhood radius $\sigma$. Results of this ablation study are summarized in Fig. 9 and Table 12 in App. C. Best performance is observed at $\sigma = 10^{-2}$. When $\sigma$ is too small ($10^{-4}$), we sample (almost) the same points and our local stability measure is not more informative than the prediction probability. When $\sigma$ is too large ($10^{-1}$), all information about the data point is lost.

• We also evaluate the Drop-Out method with varying dropout rates $p \in \{0.01, 0.1, 0.2, 0.5\}$. The correlation values between evaluated multiplicity and the stability measures for the 512-shot setting on the `Diabetes` dataset are summarized in Table 13 in App. C. Our stability measure outperforms the dropout method for all $p$ values.

• To assess the contribution of the variability term in our stability measure, we compare it to two baselines that capture only local variability: (i) absolute deviation $S_1(\mathbf{x}) = \frac{1}{k} \sum |f(\mathbf{x}_i) - f(\mathbf{x})|$, and (ii) squared deviation

*Table 3.* Correlations and runtimes on the Adult dataset (128-shot) (100 finetuned models with an overall training time of 456 mins). Train time refers to the total time required to train the models needed; Evaluation time includes inference and computation time of the method over the entire test set. Stability achieves high correlations with multiplicity metrics at lower computational cost.

| Measure | Arbit. | Pairwise Disag. | Pred. Var. | Pred. Range | Train Time | Eval. Time |
|---|---|---|---|---|---|---|
| Re-training | 1.00 | 1.00 | 1.00 | 1.00 | 456 mins | 94.7 mins |
| Pred. Prob. | 0.63 | 0.61 | 0.39 | 0.63 | 4.56 mins | 0.51 mins |
| Drop-Out | 0.79 | 0.78 | 0.70 | 0.86 | 4.56 mins | 102 mins |
| AWP | 0.65 | 0.71 | 0.55 | 0.72 | 4.56 mins | 977.6 mins |
| Stability | 0.81 | 0.96 | 0.80 | 0.93 | 4.56 mins | 19.4 mins |

$S_2(\mathbf{x}) = \frac{1}{k} \sum (f(\mathbf{x}_i) - f(\mathbf{x}))^2$. As shown in Table 10 in App. C, both alternatives yield consistently weaker correlations with multiplicity metrics, highlighting the importance of incorporating both local mean and variability terms.

• To evaluate the applicability of our stability measure beyond LoRA-based fine-tuning, we conduct an ablation using two alternative tuning strategies: Prompt Tuning (Lester et al., 2021) and Prefix Tuning (Li & Liang, 2021). As shown in Table 14 in App. C, our Stability measure continues to correlate with multiplicity, though the correlations are somewhat weaker than in the LoRA case, likely due to the limitations of these tuning methods, which are known to be less effective than LoRA in few-shot settings.

**Computational Efficiency.** We compare the computational requirements of our Stability measure against, retraining, dropout-based, Prediction probability, and Adversarial Weight Perturbation (AWP) (Hsu & Calmon, 2022) in terms of both training and evaluation runtimes. Table 3 summarizes the cumulative training time and the total evaluation time over the Adult test set. Figure 4 plots each method's total runtime and correlation with multiplicity metrics. We found AWP to be expensive for LLMs since each gradient-optimization step requires full forward passes on the test

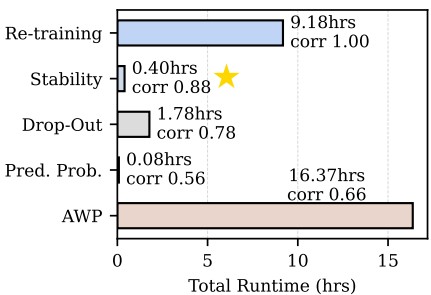

*Figure 4.* Total runtime across the `Adult` test dataset. Our proposed method (Stability) achieves significantly lower runtime compared to the re-training and baselines while maintaining strong average correlation with multiplicity evaluation metrics.

*Table 4.* Mean and std of stability values for correctly vs. incorrectly classified data points on the `Hospital` dataset. Stability achieves a larger separation between correct (0.8710) and incorrect (0.5729) data points than baselines, suggesting it is better at discriminating against unreliable predictions.

| Method | Correct | | Incorrect | |
|---|---|---|---|---|
| | Mean | Std | Mean | Std |
| Stability | 0.8710 | 0.1465 | 0.5729 | 0.1458 |
| Pred. Prob. | 0.8994 | 0.2160 | 0.7965 | 0.2256 |
| Drop-Out | 0.8190 | 0.2832 | 0.7217 | 0.1929 |

set to enforce Rashomon-set constraints, incurring heavy inference and gradient-computation costs. Dropout requires a prior check to ensure all dropped-out models (the models to be aggregated) are in the competing model set, hence our method would be more computationally efficient under the same $k$. Stability achieves the best runtime–correlation trade-off compared to baselines (see Fig. 4).

**Scalability to Large Datasets.** We evaluate our method on the `Hospital Readmission` dataset (Gardner et al., 2023)($\sim$100k data points, 50+ features). Retraining 40 models to evaluate multiplicity would takes over 5 days (3.5 hrs/model), while our method requires only a single model and a fast forward-pass sampling step. We train one model and compare Stability and baselines across correctly and incorrectly classified test points (see Table 4). Stability achieves a larger separation between correct and incorrect data points than baselines. We can also use our measure to analyze data points that are both confident and stable, or identify those that appear confident but are actually unstable. In Table 5, we grouped predictions based on their confidence and stability values. Observe that while 41% of the predictions were both confident and stable, a notable 20% of the predictions were confident but unstable, indicating that high confidence alone is not enough.

**Discussion.** Our multiplicity evaluation metrics, summarized in Table 1,6,7, reveal significant variability in model

*Table 5.* Breakdown of test predictions by confidence and stability (threshold = 0.75). 41% of predictions are both confident and stable, while a significant 20% are confident yet unstable—revealing cases where high confidence masks unreliability and underscoring the value of our stability measure.

| Pred. Prob. | Stability | % Test | Description |
|---|---|---|---|
| High ($\geq 0.75$) | High ($\geq 0.75$) | 41% | Confident & Stable |
| High ($\geq 0.75$) | Low ($< 0.75$) | 20% | Confident but Unstable |
| Low ($< 0.75$) | High ($\geq 0.75$) | 22% | Unconfident but Stable |
| Low ($< 0.75$) | Low ($< 0.75$) | 17% | Unconfident & Unstable |

predictions across different fine-tuned variants, even when they exhibit similar accuracy. This multiplicity is not captured by merely examining predicted probabilities, as predictions with high confidence can still be susceptible to multiplicity (see Fig. 3). Our stability measure, $S_{k,\sigma}(\mathbf{x}, f)$, was compared with the prediction probabilities $f(\mathbf{x})$. The results, presented in Table 2,8,9, demonstrate that our stability measure consistently shows mainly higher correlation with multiplicity metrics across all models and datasets compared to prediction probabilities and drop-out method. This indicates that $S_{k,\sigma}(\mathbf{x}, f)$ is more informative than the baselines in informing the fine-tuning multiplicity. The drop-out method is however better than the prediction probabilities alone. We hypothesize that our method is more suitable for LLMs because the embedding space of LLMs is significantly smaller than the parameter space (possibly more informative also). The drop-out method might need significantly more inferences to compete due to this.

We study the unique nature of fine-tuning multiplicity in Tabular LLMs. Marx et al. (2020); Rudin et al. (2024) argue for the necessity of measuring and reporting multiplicity to better inform predictions. Traditional methods to measure multiplicity in classical ML are impractical for LLMs due to the computational challenge of retraining several fine-tuned models (Marx et al., 2020; Hsu & Calmon, 2022; Watson-Daniels et al., 2023). Our proposed measure, which requires only the given model and leverages the embedding space to inform multiplicity, addresses this issue. This approach reduces the complexity from retraining and inference to just inference, making it more feasible to apply in practice. Although, a large $k$ (number of sampled points) may be needed for accurate stability estimation, it remains computationally more efficient than retraining multiple models. Compared to existing methods, our stability measure achieves a superior trade-off between runtime and correlation with evaluated multiplicity metrics, as shown in Figure 4. Our work provides practitioners with meaningful information about the multiplicity of predictions, which may lead them to carefully evaluate which predictions to trust and which to treat with caution. Our research has significant implications in several high-stakes applications, e.g., hiring, finance, education, etc., where inconsistent predictions can lead to distrust.

# Acknowledgements

This paper was prepared for informational purposes in part by the CDAO group of JPMorgan Chase & Co and its affiliates ("J.P. Morgan") and is not a product of the Research Department of J.P. Morgan. J.P. Morgan makes no representation and warranty whatsoever and disclaims all liability, for the completeness, accuracy or reliability of the information contained herein. This document is not intended as investment research or investment advice, or a recommendation, offer or solicitation for the purchase or sale of any security, financial instrument, financial product or service, or to be used in any way for evaluating the merits of participating in any transaction, and shall not constitute a solicitation under any jurisdiction or to any person, if such solicitation under such jurisdiction or to such person would be unlawful.

# Impact Statement

**Broader Societal Impacts.** The application of LLMs to tabular data, particularly in high-stakes domains such as finance and healthcare, presents both opportunities and risks (Bommasani et al., 2021). Our work aims to address one of the critical challenges associated with these models: the instability introduced when fine-tuning large models on small datasets. This instability, manifested as overfitting and multiplicity, can undermine the reliability of model predictions in scenarios where stability is crucial. By measuring multiplicity, our work contributes to the responsible deployment of LLMs in domains where erroneous predictions can have severe consequences (Bommasani et al., 2021; Creel & Hellman, 2022). Tabular data is central to these high-stakes domains but remains underexplored compared to text and vision (Hegselmann et al., 2023). Recent work emphasizes the need for reliable foundation models in this modality (van Breugel & van der Schaar, 2024).

Our approach also supports *regulatory compliance* by enhancing transparency and accountability in automated decision-making systems. Quantifying prediction consistency aligns with regulations such as the General Data Protection Regulation (GDPR) (Voigt, 2017) and upcoming AI legislation, which increasingly demand explainable and reliable AI models (Chamola et al., 2023). While LLMs are more computationally expensive than traditional models, our method reduces the costs of assessing multiplicity. By avoiding repeated retraining, it enhances *cost efficiency* and *minimizes environmental impact*, lowering both energy consumption and carbon footprint (Luccioni et al., 2023).

Furthermore, observing the nature of fine-tuning multiplicity in Tabular LLMs pave the way for future research into model stability. It also *facilitates continual learning* by informing the robustness of a prediction to potential model updates in a dynamic environments where data constantly evolves (Amba Hombaiah et al., 2021; Wu et al., 2024; Wang et al., 2024a). Lastly, our work could play a role in mitigating *fairwashing risks* and *explanation bias* (Black et al., 2022; Sokol et al., 2023; Rudin et al., 2024). This transparency is crucial for maintaining ethical standards and trustworthiness in AI deployment (Chamola et al., 2023).

**Limitations.** Our work provides a measure to assess fine-tuning multiplicity, but does not directly resolve this issue. Future research could focus on mitigation methods to ensure more consistent model predictions. A key constraint is the applicability to higher-dimensional datasets due to the limited context window size of current LLMs, though extending context windows is an active area of research (Peng et al., 2023; Chen et al., 2023a). Additionally, our method's performance can be sensitive to hyperparameters, such as sample size and neighborhood radius; incorrect choices may lead to an inaccurate assessment of robustness. Our approach also assumes access to the embedding space, limiting its application to open-source models. Furthermore, the bound in Theorem 1 is not directly computable. Estimating these unknowns such as $\epsilon'$ could be a direction for future work. Despite these limitations, our measure serves as a crucial step toward understanding and quantifying fine-tuning multiplicity, laying the groundwork for future advancements.

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

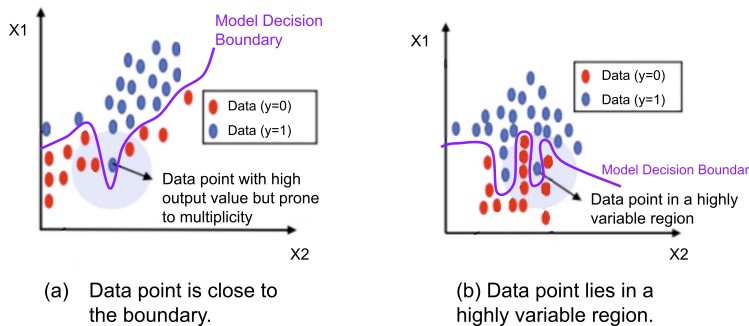

*Figure 5.* Additional motivation for our local stability measure. Our measure relies on both local variability and mean confidence as they capture synergistic aspects of prediction robustness.

## A. Additional Intuition Behind the Stability Measure

*How stability differs from existing robustness measures*: Our focus on model multiplicity distinguishes this work from traditional robustness measures, which address different aspects of model behavior such as out-of-distribution (OOD) generalization, stability under natural perturbations, and uncertainty estimation (Djolonga et al., 2020). OOD generalization typically evaluates how well a model performs on data that differs from the training distribution (e.g., classifying objects seen from novel viewpoints or in cluttered settings). This is often quantified using test datasets with altered conditions or domain shifts, and methods like domain adaptation are employed to enhance robustness. Stability under natural perturbations assesses the sensitivity of predictions and predicted probabilities to small, random changes in the input, such as Gaussian noise or image transformations. Uncertainty estimation, on the other hand, focuses on calibrating the predicted probabilities to reflect true likelihoods, often using measures like Expected Calibration Error or entropy-based metrics to evaluate how well the model quantifies confidence in its predictions. While these methods provide valuable insights into different facets of robustness, their goals differ significantly from ours.

Han et al. (2023) is more closely related to our approach, as it quantifies robustness by measuring the fraction of stable predictions within a local neighborhood. While both approaches leverage the neighborhood around a data point, the objectives diverge: Han et al. (2023) focuses on quantifying the probability of stable predictions against perturbations to evaluate robustness to noise. In contrast, our measure aims to capture the consistency of predictions (multiplicity) among competing models within the Rashomon set. Additionally, our stability measure's unique mean-variance nature further distinguishes it (see Figure 4). Unlike existing metrics, it not only accounts for the average prediction within a neighborhood but also penalizes the variability in predictions. Moreover, we provide theoretical guarantees on the consistency of predictions with high stability scores over a broad range of equally-well performing models. Recent work has also explored consistency in LLMs across repeated inference runs or under slight semantic perturbations to the input (Novikova et al., 2025; Raj et al., 2025; 2022; Nalbandyan et al., 2025).

## B. Proof of Theoretical Guarantee

**Theorem 1** (Probabilistic Guarantee). *Given a data point* $\mathbf{x}$, *a target model* $\bar{f}$ *and local stability measure* $S_{k,\sigma}(\mathbf{x}, F)$. *Under Assumption 1, and for all* $\epsilon > \epsilon'$, *where* $\epsilon' = 2(\alpha + t\sigma) + \mathcal{O}(L\sigma^2)$. *We have:*

$$\Pr\left(\bar{f}(\mathbf{x}) \geq S_{k,\sigma}(\mathbf{x}, F) - \epsilon\right) \geq 1 - \exp\left(\frac{-k\epsilon^2}{32}\right) \tag{1}$$

*Proof.* To prove Theorem 1, we begin with Lemma 1.

Assume the fine-tuned models $F$ belong to a discrete class of random variables. A specific model realization is represented as $f^i$ for $i = 1, 2, \ldots, |\mathcal{F}_\delta|$, with the complete set denoted by $\mathcal{F} = \{f^1, f^2, \ldots, f^{|\mathcal{F}|}\}$. Each model $f^i$ is selected with probability $p_i$, where $\sum_{i=1}^{|\mathcal{F}_\delta|} p_i = 1$.

**Lemma 1.** *Given* $Z_i = F(X_i) - \bar{f}(X_i) - |\bar{f}(X_i) - \bar{f}(\mathbf{x})| + |F(X_i) - F(\mathbf{x})|$ *and* $Z = \frac{1}{k}\sum_{i=1}^{k} Z_i$, *under Assumption 1,*

*for any $\tilde{\epsilon} > \epsilon' > 0$, we have:*

$$\Pr(Z \geq \epsilon' + \tilde{\epsilon}) \leq \exp\left(\frac{-k(\tilde{\epsilon} + \epsilon')^2}{32}\right). \tag{2}$$

**Lemma 2** (Hoeffding's Inequality). *For a given random variable $X_i$ such that $X_i \in [a, b]$ almost surely, and for any $\varepsilon > 0$,*

$$\Pr\left(\left|\frac{1}{k}\sum_{i=1}^{k}X_i - \mathbb{E}(X_i)\right| > \varepsilon\right) \leq 2\exp\left(-\frac{2k\varepsilon^2}{(b-a)^2}\right). \tag{3}$$

See Bentkus (2004) for detailed proof of Hoeffding's Inequality.

Since $\bar{f}(\cdot), F(\cdot) \in [0, 1]$, we have $Z_i \in [-2, 2]$. Hence, from Lemma 2, we have:

$$\Pr\left(|Z - \mathbb{E}[Z|F = f]| \geq \tilde{\epsilon} \mid F = f\right) \leq 2\exp\left(-\frac{k\tilde{\epsilon}^2}{8}\right) \tag{4}$$

Given $|\mathbb{E}[Z|F = f]| < \epsilon'$, we have $-\epsilon' < \mathbb{E}[Z|F = f] < \epsilon' \ \forall f$ (see Lemma 3). Now observe that:

$$\Pr(Z \geq \epsilon' + \tilde{\epsilon}|F = f) \overset{(a)}{\leq} \Pr(Z \geq \mathbb{E}[Z|F{=}f] + \tilde{\epsilon}|F{=}f) \leq \exp\left(\frac{-k\tilde{\epsilon}^2}{8}\right). \tag{5}$$

Here, (a) holds since $\mathbb{E}[Z|F = f] < \epsilon'$. The event on the left is a subset of that on the right. Therefore, the probability of the event $\{Z \geq \epsilon' + \tilde{\epsilon}\}$ occurring cannot be more than the probability of the event $\{Z \geq \mathbb{E}[Z|F = f] + \tilde{\epsilon}\}$ occurring.

$$\Pr(Z \geq \epsilon' + \tilde{\epsilon}) \overset{(b)}{=} \sum_i \Pr(Z \geq \epsilon' + \tilde{\epsilon}|F = f^i)\Pr(F = f^i) \tag{6}$$

$$\overset{(c)}{\leq} \exp\left(\frac{-k\tilde{\epsilon}^2}{8}\right)\sum_i \Pr(F = f^i) \tag{7}$$

$$= \exp\left(\frac{-k\tilde{\epsilon}^2}{8}\right) \tag{8}$$

$$\overset{(d)}{\leq} \exp\left(\frac{-k(\tilde{\epsilon} + \epsilon')^2}{32}\right) \tag{9}$$

Here, (b) holds from the law of total probability. Next, (c) follows from (5). Finally, (d) holds from using the inequality $4\tilde{\epsilon}^2 > (\tilde{\epsilon} + \epsilon')^2$ which holds for $\tilde{\epsilon} > \epsilon' > 0$. Setting $\epsilon = \tilde{\epsilon} + \epsilon'$.

We have:

$$\Pr\left(\frac{1}{k}\sum_{i=1}^{k}\bar{f}(X_i) \geq \frac{1}{k}\sum_{i=1}^{k}\left(F(X_i) - |F(X_i) - F(\mathbf{x})| + |\bar{f}(X_i) - \bar{f}(\mathbf{x})|\right) - \epsilon\right) \geq 1 - \exp\left(\frac{-k\epsilon^2}{32}\right). \tag{10}$$

Observe that $\bar{f}(\mathbf{x}) \geq \bar{f}(\mathbf{x}_i) - |\bar{f}(\mathbf{x}_i) - \bar{f}(\mathbf{x})|$. This applies directly from the reverse triangle inequality, i.e., for any real numbers $a$ and $b$, we have: $|a| \geq |b| - |a - b|$. Hence,

$$\bar{f}(\mathbf{x}) \geq \frac{1}{k}\sum_{i=1}^{k}(\bar{f}(X_i) - |\bar{f}(X_i) - \bar{f}(\mathbf{x})|) \tag{11}$$

Therefore, plugging (11) into (10), we have:

$$\Pr\left(\bar{f}(\mathbf{x}) \geq \frac{1}{k}\sum_{i=1}^{k}(F(X_i) - |F(X_i) - F(\mathbf{x})| + |\bar{f}(X_i) - \bar{f}(\mathbf{x})| - |\bar{f}(X_i) - \bar{f}(\mathbf{x})| - \epsilon)\right) \tag{12}$$

$$= \Pr\left(\bar{f}(\mathbf{x}) \geq \frac{1}{k}\sum_{i=1}^{k}(F(X_i) - |F(X_i) - F(\mathbf{x})|) - \epsilon\right) \geq 1 - \exp\left(\frac{-k\epsilon^2}{32}\right). \tag{13}$$

Given $S_{k,\sigma}(\mathbf{x}, F) = \frac{1}{k} \sum_{i=1}^{k} (F(X_i) - |F(\mathbf{x}) - F(X_i)|)$, we have:

$$\Pr\left(\bar{f}(\mathbf{x}) \geq S_{k,\sigma}(\mathbf{x}, F) - \epsilon\right) \geq 1 - \exp\left(\frac{-k\epsilon^2}{32}\right). \tag{14}$$

Using Lemma 3, we show that $\mathbb{E}[Z|F = f] \leq \epsilon'$ which completes the proof.

**Lemma 3.** *Let $F, \bar{f}$ be twice differentiable functions with Hessians bounded by $L$, i.e., $\|\nabla^2 F(\mathbf{x})\|, \|\nabla^2 \bar{f}(\mathbf{x})\| \leq L$ for all $\mathbf{x}$. Let $\mathbb{E}_{X_i}[\|F(X_i) - \bar{f}(X_i)\| | F = f] \leq \alpha$ and $\mathbb{E}_{X_i}[\|\nabla(F - \bar{f})(X_i)\| | F = f] \leq t$, where $X_i$ is drawn uniformly from distribution over the hypersphere $B(\mathbf{x}, \sigma)$. If $Z_i = \left[F(X_i) - \bar{f}(X_i)\right] - \left|\bar{f}(X_i) - \bar{f}(\mathbf{x})\right| + \left|F(X_i) - F(\mathbf{x})\right|$, then,*

$$\mathbb{E}[Z_i | F = f] \leq \alpha + t\sigma + \underbrace{\mathcal{O}(L\sigma^2)}_{\text{Hessian Error}}, \tag{15}$$

*Proof.* We first bound $\mathbb{E}[F(X_i) - \bar{f}(X_i)]$:

Since $\mathbb{E}[|F(X_i) - \bar{f}(X_i)|] \leq \alpha$ for all it follows that $\mathbb{E}[F(X_i) - \bar{f}(X_i)] \leq \alpha$.

Next, we bound $\mathbb{E}\left[|F(X_i) - F(\mathbf{x})| - |\bar{f}(X_i) - \bar{f}(\mathbf{x})|\right]$:

Expand $F(X_i)$ and $\bar{f}(X_i)$ around $\mathbf{x}$ using Taylor's expansion:

$$\begin{aligned}
F(\mathbf{x}) &= F(X_i) + \nabla F(X_i)^T (\mathbf{x} - X_i) + \frac{1}{2}(\mathbf{x} - X_i)^T \nabla^2 F(\xi_F)(\mathbf{x} - X_i), \\
\bar{f}(\mathbf{x}) &= \bar{f}(X_i) + \nabla \bar{f}(X_i)^T (\mathbf{x} - X_i) + \frac{1}{2}(\mathbf{x} - X_i)^T \nabla^2 \bar{f}(\xi_{\bar{f}})(\mathbf{x} - X_i),
\end{aligned} \tag{16}$$

for some $\xi_F, \xi_{\bar{f}} \in B(\mathbf{x}, \sigma)$. The absolute differences become:

$$\begin{aligned}
|F(X_i) - F(\mathbf{x})| &\leq \left|\nabla F(X_i)^T (\mathbf{x} - X_i)\right| + \frac{1}{2}L\|X_i - \mathbf{x}\|^2, \\
|\bar{f}(X_i) - \bar{f}(\mathbf{x})| &\geq \left|\nabla \bar{f}(X_i)^T (\mathbf{x} - X_i)\right| - \frac{1}{2}L\|X_i - \mathbf{x}\|^2.
\end{aligned} \tag{17}$$

Subtracting these:

$$|F(X_i) - F(\mathbf{x})| - |\bar{f}(X_i) - \bar{f}(\mathbf{x})| \leq \left|\nabla F(X_i)^T (\mathbf{x} - X_i)\right| - \left|\nabla \bar{f}(X_i)^T (\mathbf{x} - X_i)\right| + L\|X_i - \mathbf{x}\|^2. \tag{18}$$

Using the reverse triangle inequality $|a| - |b| \leq |a - b|$:

$$|F(X_i) - F(\mathbf{x})| - |\bar{f}(X_i) - \bar{f}(\mathbf{x})| \leq \left|\nabla(F - \bar{f})(X_i)^T (\mathbf{x} - X_i)\right| + L\|X_i - \mathbf{x}\|^2. \tag{19}$$

Taking Expectation:

$$\mathbb{E}\left[\left|\nabla(F - \bar{f})(X_i)^T (\mathbf{x} - X_i)\right|\right] \leq \mathbb{E}[\|\nabla(F - \bar{f})(X_i)\| \|X_i - \mathbf{x}\|] \leq \mathbb{E}[\|\nabla(F - \bar{f})(X_i)\| \sigma \leq t\sigma \tag{20}$$

Hessian Term: $\mathbb{E}\left[L\|X_i - \mathbf{x}\|^2\right] = L \cdot \mathbb{E}[\|X_i - \mathbf{x}\|^2] = \mathcal{O}(L\sigma^2)$.

Combining the inequalities, we have:

$$\mathbb{E}[Z_i | F = f] \leq \alpha + t\sigma + \mathcal{O}(L\sigma^2). \tag{21}$$

$\square$

$\square$

# C. Expanded Experimental Section

## C.1. Relevant Definition

**Definition 6** (Spearman Correlation). *Spearman's correlation, Spearman$(X, Y)$, measures the strength and direction of a monotonic relationship between two variables. It is the Pearson correlation coefficient of their ranked values. Given $n$ pairs $(X_i, Y_i)$, it is computed as:*

$$Spearman(X, Y) = 1 - \frac{6 \sum_{i=1}^{n} d_i^2}{n(n^2 - 1)} = \frac{cov(rank(X), rank(Y))}{\sigma_{rank(X)} \sigma_{rank(Y)}},$$

*where $d_i$ is the difference between the ranks of $X_i$ and $Y_i$. The value ranges from $-1$ (perfect negative monotonicity) to $1$ (perfect positive monotonicity), with $0$ indicating no monotonic relationship.*

## C.2. Dataset Details

This section provides detailed descriptions of the datasets used in our experiments.

**Adult** dataset (Becker & Kohavi, 1996), also known as the "Census Income" dataset, is used for predicting whether an individual earns more than $50,000 annually based on various demographic attributes. It consists of 48,842 instances with 14 attributes, including age, work class, education, marital status, occupation, relationship, race, sex, capital gain, capital loss, hours per week, and native country. The dataset is commonly used in classification tasks.

**German Credit** dataset (Hofmann, 1994) is used for credit risk evaluation. It consists of $1,000$ instances with $20$ attributes, which include personal information, credit history, and loan attributes. The target variable indicates whether the credit is good or bad. This dataset is often used for binary classification problems and helps in understanding the factors affecting creditworthiness. The dataset is commonly used in classification tasks.

**Diabetes** dataset (Kahn) is used for predicting the onset of diabetes based on diagnostic measurements. It contains 768 instances with 8 attributes, including the number of pregnancies, glucose concentration, blood pressure, skin thickness, insulin level, body mass index (BMI), diabetes pedigree function, and age. The target variable indicates whether the individual has diabetes. The dataset is commonly used in classification tasks.

**Bank** dataset (Moro et al., 2014) is used for predicting whether a client will subscribe to a term deposit based on data from direct marketing campaigns of a Portuguese bank. It includes 45,211 instances in the training set and 18 attributes, such as age, job type, marital status, education, credit balance, housing loan status, and contact details from the marketing campaigns. The target variable indicates whether the client subscribed to the term deposit. This dataset is commonly used in binary classification tasks.

**Heart** dataset (Detrano et al., 1989) contains data from four different hospitals. It includes 918 patients, each represented by 11 clinical variables, with the task being a binary classification of coronary artery disease. Among the patients, 508 are labeled positive for the condition.

**Car** dataset (Kadra et al., 2021) contains entries describing various cars characterized by six attributes. The task is a classification problem aimed at evaluating the state of each car. The dataset comprises 1,728 examples.

**Hospital Remission** dataset (Gardner et al., 2023) contains 99,493 inpatient encounters from 130 U.S. hospitals (1999–2008) that involve diabetic patients. The dataset provides 50+ attributes aimed at predicting whether a patient will be readmitted within 30 days after discharge. It is commonly used for binary classification tasks assessing post-discharge risk and care quality.

## C.3. Experimental Setup

Our experiments were carried out using the BIGSCIENCE T0 and Google Flan T5 models fine-tuned on several datasets. The number of shots was set to $64, 128$, and $512$ for each dataset. To evaluate multiplicity and local stability, we fine-tuned 40 models with different random seeds for each dataset and recorded their predictions. The training process involved setting the batch size to 2 for smaller training sizes and 8 for larger sizes. The learning rate was set to 0.003. For each dataset, we determined the number of training steps adaptively based on the number of shots, ensuring sufficient iterations for model convergence. Specifically, the training steps were calculated as $20 \times$ (number of shots/batch size). All experiments were performed on 2 NVIDIA RTX A4500 and 4 NVIDIA RTX 6000 GPUs. To ensure reproducibility and robustness of the

results, different random seeds (i.e., 2, 4, 8, etc) were used for each fine-tuning iteration. For fine-tuning with LoRA we use a rank of 4. Given the infeasibility of computing the exact size of $|\mathcal{F}_\delta|$ due to its potentially vast model space, we employ an expensive sampling approach, i.e., fine-tuning with various seeds. We select a finite number of models from $\mathcal{F}_\delta$ for practical evaluation, allowing us to evaluate the multiplicity metrics. It is very computationally expensive to fine-tune several models to evaluate multiplicity. This motivates the need for a measure to quantify stability given one model.

### C.4. Expanded Results

This section presents a broader set of experimental results. We evaluate multiplicity across several datasets using both the BIGSCIENCE T0 model fine-tuned with LoRA (see Table 6) and the FLAN-T5 model fine-tuned using the Tfew recipe (see Table 7). Additionally, Figures 6, 7, and 8 visualize the evaluated multiplicity versus the stability measure for `Bank`, `Diabetes`, and `German Credit` datasets respectively. We also report the correlation between the stability measure and various multiplicity evaluation metrics for BIGSCIENCE T0 model fine-tuned using Tfew recipe (see Table 8) and FLAN-T5 model fine-tuned using Tfew recipe (see Table 9).

*Table 6.* Multiplicity Evaluation Metrics for Different Datasets and Number of Shots. Evaluated on 40 fine-tuned **BIGSCIENCE T0** models on **LoRA** using different random seeds. Multiplicity observed in predictions across different fine-tuned model, even when models exhibit similar accuracy (in this setting $\delta = 0.02$).

| Dataset | No. Shots | Multiplicity Evaluation Metrics (BIGSCIENCE T0) | | | | | |
| --- | --- | --- | --- | --- | --- | --- | --- |
| | | **Arbitrariness** | **Discrepancy** | **Avg. Pairwise Disagreement** | **Avg. Pred. Variance** | **Avg. Pred. Range** | **Avg. Model Accuracy** |
| Adult | 64 | 11% | 6% | 9% | 0.01 | 0.11 | 83% |
| | 128 | 10% | 9% | 6% | 0.01 | 0.10 | 84% |
| | 512 | 11% | 3% | 10% | 0.01 | 0.12 | 85% |
| German | 64 | 19% | 10% | 6% | 0.04 | 0.40 | 70% |
| | 128 | 17% | 11% | 6% | 0.01 | 0.16 | 71% |
| | 512 | 21% | 14% | 8% | 0.03 | 0.26 | 72% |
| Diabetes | 64 | 20% | 13% | 11% | 0.04 | 0.21 | 70% |
| | 128 | 16% | 14% | 11% | 0.08 | 0.14 | 73% |
| | 512 | 19% | 13% | 11% | 0.04 | 0.17 | 76% |
| Bank | 64 | 13% | 9% | 7% | 0.01 | 0.28 | 66% |
| | 128 | 14% | 9% | 7% | 0.03 | 0.21 | 73% |
| | 512 | 14% | 8% | 7% | 0.03 | 0.22 | 78% |

*Table 7.* Evaluated Multiplicity for Different Datasets and Number of Shots. Evaluated on 40 fine-tuned **FLAN-T5** models using **Tfew** recipe with different random seeds. Multiplicity observed in predictions across different fine-tuned models, even when models exhibit similar accuracy (in this setting $\delta = 0.02$). The accuracy of FLAN T5 model on the dataset is less than the BIGSCIENCE T0 model observed in Table 1.

| Dataset | No. Shots | Multiplicity Evaluation Metrics (FLAN-T5) | | | | | |
| --- | --- | --- | --- | --- | --- | --- | --- |
| | | **Arbitrariness** | **Discrepancy** | **Avg. Pairwise Disagreement** | **Avg. Pred. Variance** | **Avg. Pred. Range** | **Avg. Model Accuracy** |
| Adult | 64 | 13.96% | 6.93% | 5.05% | 0.010 | 0.139 | 74.25% |
| | 128 | 8.81% | 3.84% | 3.39% | 0.008 | 0.091 | 77.50% |
| | 512 | 12.02% | 5.71% | 4.49% | 0.012 | 0.123 | 79.17% |
| German | 64 | 18.50% | 11.00% | 6.19% | 0.015 | 0.194 | 64.85% |
| | 128 | 30.00% | 13.50% | 10.47% | 0.031 | 0.287 | 69.25% |
| | 512 | 35.50% | 16.50% | 12.88% | 0.041 | 0.362 | 69.40% |
| Diabetes | 64 | 15.58% | 7.79% | 6.23% | 0.016 | 0.170 | 68.18% |
| | 128 | 11.69% | 5.84% | 4.81% | 0.012 | 0.129 | 59.29% |
| | 512 | 21.43% | 9.74% | 7.37% | 0.022 | 0.207 | 69.55% |
| Bank | 64 | 12.86% | 7.46% | 4.69% | 0.003 | 0.125 | 66.96% |
| | 128 | 17.95% | 6.90% | 6.59% | 0.006 | 0.165 | 65.94% |
| | 512 | 17.17% | 6.61% | 6.24% | 0.017 | 0.173 | 79.40% |

*Table 8.* This table reports the Spearman correlation between the stability measure, predicted probabilities, and the drop-out method with various multiplicity evaluation metrics for different numbers of shots on several datasets (**BIGSCIENCE T0 fine-tuned using Tfew recipe**). In most cases, the stability measure $S_{k,\sigma}(\mathbf{x}, f)$ shows a higher correlation with these multiplicity measures compared to predicted probabilities and drop-out, indicating that the stability measure $S_{k,\sigma}(\mathbf{x}, f)$ better informs about the multiplicity than the other measures do. The dropout method performing better than naive predicted probability.

| Dataset | Number of Shots | Measure | Arbitrariness | Pairwise Disagreement | Prediction Variance | Prediction Range |
|---|---|---|---|---|---|---|
| Adult | 64 | Pred. Prob. | 0.67 | 0.66 | 0.50 | 0.62 |
| | | Drop-Out | 0.83 | 0.78 | 0.81 | 0.87 |
| | | Stability | **0.95** | **0.90** | **0.91** | **0.89** |
| | 128 | Pred. Prob. | 0.67 | 0.62 | 0.30 | 0.54 |
| | | Drop-Out | 0.74 | 0.83 | 0.69 | 0.81 |
| | | Stability | **0.80** | **0.96** | **0.84** | **0.91** |
| | 512 | Pred. Prob. | 0.70 | 0.69 | 0.56 | 0.72 |
| | | Drop-Out | 0.78 | 0.78 | 0.88 | 0.88 |
| | | Stability | **0.90** | **0.86** | **0.93** | **0.92** |
| German Credit | 64 | Pred. Prob. | **0.99** | **0.99** | 0.80 | 0.79 |
| | | Drop-Out | 0.73 | 0.71 | 0.82 | 0.76 |
| | | Stability | 0.95 | 0.95 | **0.98** | **0.84** |
| | 128 | Pred. Prob. | **0.57** | **0.57** | 0.86 | 0.86 |
| | | Drop-Out | 0.50 | 0.56 | 0.74 | 0.84 |
| | | Stability | 0.54 | 0.54 | **0.87** | **0.87** |
| | 512 | Pred. Prob. | 0.54 | 0.56 | 0.83 | 0.82 |
| | | Drop-Out | **0.69** | **0.67** | 0.72 | 0.65 |
| | | Stability | 0.59 | 0.60 | **0.87** | **0.86** |
| Diabetes | 64 | Pred. Prob. | 0.03 | 0.38 | 0.04 | 0.08 |
| | | Drop-Out | 0.30 | 0.19 | **0.54** | **0.46** |
| | | Stability | **0.45** | **0.51** | 0.31 | 0.23 |
| | 128 | Pred. Prob. | 0.88 | 0.93 | **0.93** | **0.95** |
| | | Drop-Out | 0.89 | 0.92 | 0.92 | 0.94 |
| | | Stability | **0.92** | **0.95** | 0.93 | 0.95 |
| | 512 | Pred. Prob. | 0.21 | 0.23 | 0.24 | 0.30 |
| | | Drop-Out | 0.74 | 0.83 | **0.75** | **0.74** |
| | | Stability | **0.80** | **0.89** | 0.74 | 0.68 |
| Bank | 64 | Pred. Prob. | 0.70 | 0.69 | 0.56 | 0.74 |
| | | Drop-Out | 0.79 | 0.77 | 0.77 | **0.80** |
| | | Stability | **0.83** | **0.78** | **0.81** | 0.80 |
| | 128 | Pred. Prob. | 0.54 | 0.57 | 0.73 | 0.62 |
| | | Drop-Out | 0.62 | 0.70 | 0.75 | 0.51 |
| | | Stability | **0.79** | **0.84** | **0.87** | **0.86** |
| | 512 | Pred. Prob. | 0.71 | 0.68 | 0.81 | 0.76 |
| | | Drop-Out | 0.90 | 0.89 | 0.87 | 0.84 |
| | | Stability | **0.91** | **0.92** | **0.91** | **0.87** |
| Heart | 64 | Pred. Prob. | 0.70 | 0.21 | 0.30 | 0.69 |
| | | Drop-Out | 0.56 | 0.48 | 0.54 | 0.56 |
| | | Stability | **0.98** | **0.86** | **0.98** | **0.98** |
| | 128 | Pred. Prob. | 0.61 | 0.46 | 0.50 | 0.26 |
| | | Drop-Out | 0.64 | 0.76 | 0.74 | 0.83 |
| | | Stability | **0.89** | **0.90** | **0.97** | **0.87** |
| | 512 | Pred. Prob. | 0.80 | 0.65 | 0.48 | 0.35 |
| | | Drop-Out | **0.94** | 0.90 | **0.90** | 0.94 |
| | | Stability | 0.89 | **0.95** | 0.86 | **0.95** |
| Car | 64 | Pred. Prob. | 0.83 | **0.83** | 0.40 | 0.83 |
| | | Drop-Out | **0.85** | **0.83** | **0.96** | **0.97** |
| | | Stability | 0.76 | 0.69 | 0.86 | 0.75 |
| | 128 | Pred. Prob. | 0.56 | 0.26 | 0.29 | 0.01 |
| | | Drop-Out | 0.63 | 0.66 | 0.57 | 0.52 |
| | | Stability | **0.97** | **0.91** | **0.93** | **0.94** |
| | 512 | Pred. Prob. | 0.91 | 0.94 | 0.72 | 0.86 |
| | | Drop-Out | **0.98** | **0.96** | **0.95** | **0.93** |
| | | Stability | 0.68 | 0.59 | 0.56 | 0.67 |

*Table 9.* This table reports the Spearman correlation between the predicted probabilities, drop-out method, and the stability measure with various multiplicity evaluation metrics for different numbers of shots on several datasets (**Flan T5 model fine-tuned using Tfew recipe**). In most cases, the stability measure shows a higher correlation with these multiplicity measures compared to predicted probabilities and drop-out, indicating that the stability measure better informs about the multiplicity than the other measures do. The dropout method performs competitively in some cases.

| Dataset | Number of Shots | Measure | Arbitrariness | Pairwise Disagreement | Prediction Variance | Prediction Range |
|---|---|---|---|---|---|---|
| Adult | 64 | Pred. Prob. | 0.62 | 0.67 | **0.72** | 0.56 |
| | | Drop-Out | 0.60 | 0.65 | 0.67 | 0.57 |
| | | Stability | **0.63** | **0.72** | **0.72** | **0.60** |
| | 128 | Pred. Prob. | 0.75 | 0.74 | 0.65 | 0.75 |
| | | Drop-Out | 0.85 | 0.78 | 0.83 | 0.75 |
| | | Stability | **0.88** | **0.90** | **0.84** | **0.79** |
| | 512 | Pred. Prob. | 0.78 | 0.68 | 0.42 | 0.45 |
| | | Drop-Out | 0.78 | 0.78 | 0.42 | 0.45 |
| | | Stability | **0.79** | **0.71** | **0.78** | **0.68** |
| German Credit | 64 | Pred. Prob. | 0.27 | 0.04 | 0.27 | 0.17 |
| | | Drop-Out | 0.73 | 0.45 | 0.60 | 0.17 |
| | | Stability | **0.77** | **0.67** | **0.78** | **0.76** |
| | 128 | Pred. Prob. | 0.85 | 0.76 | 0.85 | 0.91 |
| | | Drop-Out | 0.86 | 0.91 | 0.85 | 0.91 |
| | | Stability | **0.89** | **0.91** | **0.89** | **0.92** |
| | 512 | Pred. Prob. | 0.42 | 0.29 | 0.27 | 0.19 |
| | | Drop-Out | 0.43 | 0.36 | 0.28 | 0.33 |
| | | Stability | **0.61** | **0.60** | **0.67** | **0.69** |
| Diabetes | 64 | Pred. Prob. | 0.09 | 0.04 | 0.27 | 0.23 |
| | | Drop-Out | 0.24 | 0.41 | **0.54** | **0.50** |
| | | Stability | **0.27** | **0.55** | 0.31 | 0.25 |
| | 128 | Pred. Prob. | 0.16 | 0.06 | 0.17 | 0.16 |
| | | Drop-Out | 0.46 | 0.55 | **0.54** | **0.63** |
| | | Stability | **0.52** | **0.57** | 0.44 | 0.52 |
| | 512 | Pred. Prob. | 0.61 | 0.35 | 0.12 | 0.19 |
| | | Drop-Out | 0.71 | 0.42 | **0.42** | **0.51** |
| | | Stability | **0.79** | **0.40** | 0.39 | 0.40 |
| Bank | 64 | Pred. Prob. | 0.26 | 0.04 | 0.27 | 0.17 |
| | | Drop-Out | 0.24 | 0.60 | 0.60 | **0.60** |
| | | Stability | **0.77** | **0.67** | **0.78** | 0.76 |
| | 128 | Pred. Prob. | 0.45 | 0.54 | 0.73 | 0.62 |
| | | Drop-Out | 0.62 | 0.70 | 0.75 | **0.82** |
| | | Stability | **0.89** | **0.71** | **0.78** | 0.84 |
| | 512 | Pred. Prob. | 0.42 | 0.29 | 0.27 | 0.11 |
| | | Drop-Out | 0.44 | 0.29 | **0.37** | **0.43** |
| | | Stability | **0.61** | **0.60** | 0.30 | 0.380 |

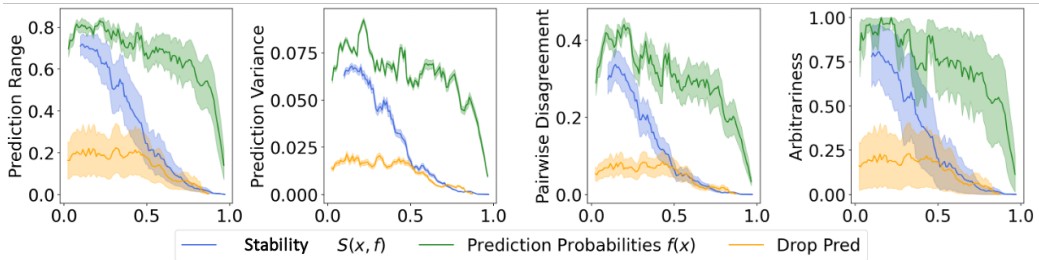

*Figure 6.* Evaluated multiplicity (assessed on 40 retrained models) versus our stability measure (evaluated on one model) for the **512-shot** setting on the **Bank dataset**. The plots demonstrate that high stability values correspond to low multiplicity across various multiplicity evaluation metrics. Predictive probabilities and Drop-Out not providing any providing any useful insight into multiplicity.

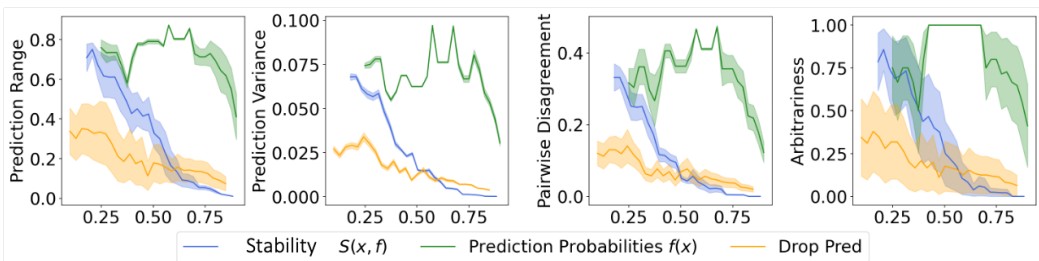

*Figure 7.* Evaluated multiplicity (assessed on 40 retrained models) versus our stability measure (evaluated on one model) for the **512-shot** setting on the **Diabetes dataset**. The plots demonstrate that high stability values correspond to low multiplicity across various multiplicity evaluation metrics. Predictive probabilities not providing any providing any useful insight about multiplicity. The drop-out method performs better than predictive probabilities but still worse than stability.

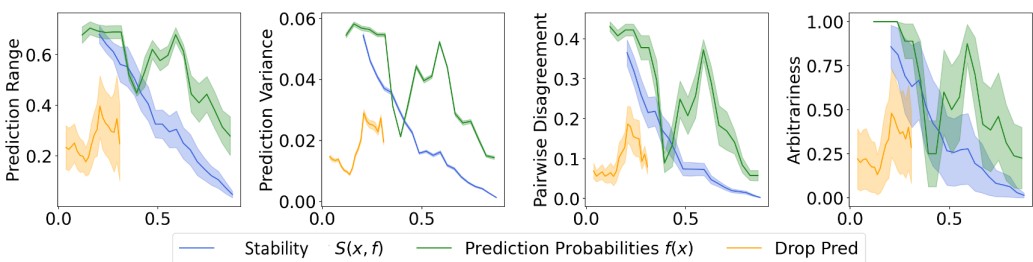

*Figure 8.* Evaluated multiplicity (assessed on 40 retrained models) versus our stability measure (evaluated on one model) for the **512-shot** setting on the **German Credit dataset**. The plots demonstrate that high stability values correspond to low multiplicity across various multiplicity evaluation metrics. In this setting Prediction probability is performing competitively. But generally stability measure provides better insight into the multiplicity of predictions compared to the predicted probabilities. The drop-out method is performing significantly worse than the other two measures.

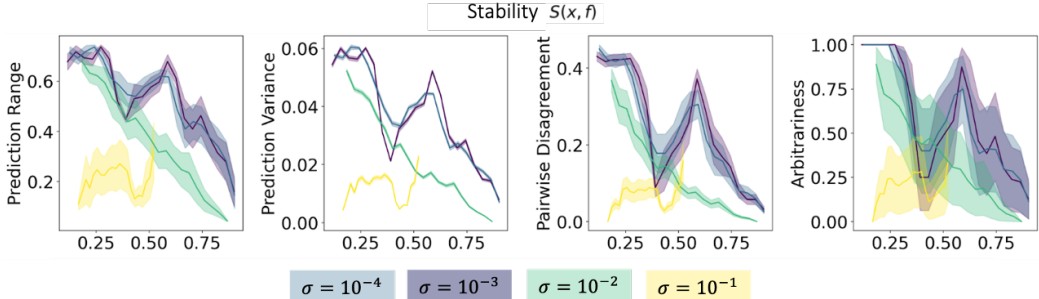

*Figure 9.* **Ablation study on different** $\sigma$ **values**: The chosen value of $\sigma = 0.01$ yields the best performance across all evaluation metrics. Smaller values of $\sigma$ (e.g., $\sigma = 10^{-4}$) result in perturbations that are too close to the original data points, leading to similar outcomes as prediction probability alone, as the sampled points are nearly identical. On the other hand, larger values (e.g., $\sigma = 10^{-2}$) produce overly noisy perturbations, rendering the results uninformative.

## C.5. Expanded Ablations

We include the complete ablation results referenced in the main paper. Table 11 presents the results of varying the sample size $k$. Table 12 and Figures 9 examines the effect of different values of $\sigma$ on the stability measure. Additionally, Table 13 compares the stability measure with the drop-out method with different drop-out rates $p$. Finally, Table 10 compares the stability measure with variability-based alternatives. Table 14 compares performance under LoRA, Prompt Tuning, and Prefix Tuning on the `Bank` dataset.

*Table 10.* Correlation between the proposed consistency measures and various multiplicity evaluation metrics. New additional measures: $S_1(\mathbf{x}) = \frac{1}{k} \sum |f(\mathbf{x}_i) - f(\mathbf{x})|$ (absolute variability). $S_2(\mathbf{x}) = \frac{1}{k} \sum (f(\mathbf{x}_i) - f(\mathbf{x}))^2$ (squared variability). The proposed Stability measure outperforms both dropout-based method and purely variability-based alternatives

| Dataset | Number of Shots | Measure | Arbitrariness | Pairwise Disagreement | Prediction Variance | Prediction Range |
|---------|-----------------|---------|---------------|-----------------------|---------------------|------------------|
| Adult | 128 | Pred. Prob. | 0.67 | 0.62 | 0.30 | 0.54 |
| | | Drop-Out | 0.74 | 0.83 | 0.69 | 0.81 |
| | | $S_1(\mathbf{x})$ | 0.70 | 0.65 | 0.63 | 0.72 |
| | | $S_2(\mathbf{x})$ | 0.70 | 0.64 | 0.60 | 0.73 |
| | | Stability | **0.80** | **0.96** | **0.84** | **0.91** |

*Table 11.* **Ablation study on different** $k$ **values**: Correlation between our stability measure (evaluated on a single model) and various measures of multiplicity for different sample sizes $k$ on the `Diabetes` dataset (T0 model). We observe better performance with increasing $k$ as suggested by our theoretical results. Larger sample size $k$ values are advantageous, as they ensure that the guarantees hold with high probability. However, computational cost of model inference (forward pass) increases.

| $k$ | Prediction Range | Prediction Variance | Pairwise Disagreement | Arbitrariness |
|-----|------------------|---------------------|-----------------------|---------------|
| 2 | 0.77 | 0.77 | 0.53 | 0.52 |
| 5 | 0.82 | 0.83 | 0.56 | 0.55 |
| 10 | 0.87 | 0.87 | 0.62 | 0.61 |
| 20 | **0.89** | **0.88** | **0.70** | **0.79** |

*Table 12.* **Ablation study on different $\sigma$ values**: Correlation between our stability measure (evaluated on one model) and various evaluation measures for different values of $\sigma$ and evaluated multiplicity for `Diabetes` dataset and 128-shot case (T0 model). Best performance observed when $\sigma = 10^{-2}$. To guide the choice of $\sigma$, one could consider the spread of training data points in the embedding space (e.g., we use a value equivalent to 10% of the variance of the training data). For all our experiments, we used a fixed value of 0.01, which consistently worked well across different datasets and experiments. When $\sigma$ is too small, we basically sample (almost) the same points and our stability measure is not more informative than the prediction probability. When $\sigma$ is too large, one loses all information about the data point.

| $\sigma$ | Prediction Range | Prediction Variance | Pairwise Disagreement | Arbitrariness |
|---|---|---|---|---|
| $10^{-4}$ | 0.82 | 0.83 | 0.84 | 0.80 |
| $10^{-3}$ | 0.91 | 0.92 | 0.90 | 0.86 |
| $10^{-2}$ | **0.95** | **0.93** | **0.95** | **0.92** |
| $10^{-1}$ | 0.10 | 0.08 | 0.33 | 0.23 |

*Table 13.* This table reports the correlation between the stability measure and various evaluated multiplicity for the 512-shot setting on the `Diabetes` dataset. The stability measure $S_{k,\sigma}(x, f)$ shows a higher correlation with multiplicity compared to predicted probabilities and drop-out and ensemble method, indicating that the stability measure $S_{k,\sigma}(x, f)$ better informs multiplicity than the other measures.

| Method | Arbitrariness | Pairwise Disagreement | Prediction Variance | Prediction Range |
|---|---|---|---|---|
| Pred. Prob. | 0.21 | 0.23 | 0.24 | 0.30 |
| drop-out $p = 0.01$ | 0.21 | 0.23 | 0.27 | 0.28 |
| drop-out $p = 0.1$ | 0.62 | 0.61 | 0.59 | 0.64 |
| drop-out $p = 0.2$ | 0.74 | 0.36 | 0.53 | 0.54 |
| drop-out $p = 0.5$ | 0.16 | 0.17 | 0.18 | 0.16 |
| Stability | **0.80** | **0.89** | **0.74** | **0.68** |

*Table 14.* We compare the correlation in our default LoRA setting against Prompt Tuning and Prefix Tuning (`Bank` dataset) to assess the generalizability of our method beyond LoRA. Although the stability measure achieves the highest correlations under LoRA, it still provides meaningful signals under Prompt and Prefix tuning.

| Dataset | Measure | Arbitrariness | Pairwise Disagreement | Prediction Variance | Prediction Range |
|---|---|---|---|---|---|
| | Pred. Prob. (LoRA) | **0.54** | **0.57** | **0.73** | **0.62** |
| | Pred. Prob. (Prompt Tuning) | 0.50 | 0.48 | 0.61 | 0.55 |
| | Pred. Prob. (Prefix Tuning) | 0.52 | 0.49 | 0.59 | 0.51 |
| | Drop-Out (LoRA) | **0.62** | **0.70** | **0.75** | **0.51** |
| `Bank` | Drop-Out (Prompt Tuning) | 0.48 | 0.53 | 0.60 | 0.49 |
| | Drop-Out (Prefix Tuning) | 0.55 | 0.50 | 0.58 | 0.46 |
| | Stability (LoRA) | **0.79** | **0.84** | **0.87** | **0.86** |
| | Stability (Prompt Tuning) | 0.63 | 0.60 | 0.58 | 0.61 |
| | Stability (Prefix Tuning) | 0.59 | 0.62 | 0.60 | 0.57 |

