# OpenReview forum: "Quantifying Prediction Consistency Under Fine-tuning Multiplicity in Tabular LLMs"
_ICML.cc/2025/Conference — ICML 2025 poster_

### Official Review · Reviewer_3rKa · 2025-03-14

**Overall Recommendation:** 3

**Summary:**

The paper proposes a proxy measure of multiplicity defined as the difference in average prediction of the model on a hypersphere of radius $\sigma$ around the point $x$ and the mean absolute difference between predictions on the hypersphere and the point $x$. The paper shows that this measure strongly correlates with multiple other measures of multiplicity over the randomness of LoRA finetuning, but without the need to retrain models.

## Update after rebuttal

The rebuttal has largely addressed the concerns of my initial review, thus I have increased my score. I still would not recommend naming the concept as either "stability" or "consistency", as the latter has also been used in the literature on predictive multiplicity to mean something else. Consider something technically descriptive along the lines of "Confidence-Variability Discrepancy" or "Local Confidence Consistency", which does not imply immediate interpretation as prediction stability.

**Claims And Evidence:**

The paper has two main claims. First, that the proposed measure provides a high-probability lower bound on the confidence of the prediction  of a finetuned model. This claim comes with a proof and depends on a set of restrictive assumptions, but is not directly evaluated. It would be helpful to evaluate the claim on a toy dataset, e.g., in its in-expectation version which would be easier to estimate in practice.

The second core claim is that the proposed stability measure highly correlates with most other multiplicity measures without requiring to retrain models. This is supported by an empirical comparison to a set of existing metrics, when using 40 re-trainings. However, the paper lacks a comparison to adversarial weight perturbation (see essential references for more details) which, although is more expensive than inference, is still significantly cheaper than retraining.

**Essential References Not Discussed:**

- The discussion of the [Rashomon capacity](https://arxiv.org/abs/2206.01295) is incomplete. L418 seems to suggest that the methods proposed therein rely on retraining, but they rely on adversarial weight perturbation, which is significantly computationally cheaper than retraining. The paper lacks comparison to an AWP based evaluation of either prediction range or capacity metric.
- Prediction variance is not a new metric as L179 seems to suggest. It was previously [studied](https://arxiv.org/abs/2302.14517) and shown to be proportional to pairwise disagreement in the case that the hard class predictions are used.

**Ethical Review Concerns:**

There is no need for a full ethics review, but I would just like to point out that the Diabetes dataset has been withdrawn, and is not recommended for non-diabetes research.

**Experimental Designs Or Analyses:**

40 re-trainings seems quite low for multiplicity measures based on retraining, so it is likely that the numbers have very high variance. It would be helpful to have more re-trainings at least for some settings. [Recent work](https://arxiv.org/abs/2302.14517) suggests in the order of thousands re-trainings is needed to estimate, e.g., prediction variance and disagreement accurately.

**Methods And Evaluation Criteria:**

The method and experimental settings are reasonable and appropriate. However, as the proposed measure does not have a clear operational meaning (see other strengths and weaknesses for details), and seems only useful insofar it correlates with other operational measures, an important question that is not answered by the paper is whether the metric is applicable beyond LoRA. Thus, it would be helpful to see the whether the metric is useful outside of LoRA, e.g., full finetuning or other adapter methods.

**Other Comments Or Suggestions:**

- Table 6 in the appendix has the method rows in a different order than the other tables (stability is last)
- Stability is used for distinctly different notions in the literature on multiplicity and learning theory. I would suggest picking a different name, especially considering that the intepretation (Theorem 3.3) is not really stability but rather something like a test for high confidence.
- The focus on Tabular LLM finetuning seems a bit odd. The proposed methods should be applicable to other standard settings such as evaluating multiplicity for question answering (think MedMCQA) after LoRA finetuning.
- It would be helpful to expand to break down the "assumption" into an actual list of assumptions, e.g., unbiasedness, bounded Hessian, etc, with a clear justification for each.
- Watson-Daniels is not cited for prediction range in L140.

**Other Strengths And Weaknesses:**

The proposed method, although significantly less expensive to compute than retraining-based measures of multiplicity, has significant weaknesses:
- Arbitrary choice of $\sigma$. First, it seems that there is no principled way to pick $\sigma$. The choice will likely depend on the dataset, and assuming that the use case of the measure is not to evaluate other standard metrics, then we cannot use other measures as benchmarks to select $\sigma$. How to choose $\sigma$ in practice? E.g., there exist [principled ways to choose the Rashomon set parameter](https://www.researchgate.net/profile/Lucas-Paes-7/publication/373307229_On_the_Inevitability_of_the_Rashomon_Effect/links/671185ff069cb92a811a550e/On-the-Inevitability-of-the-Rashomon-Effect.pdf).
- Lack of operational meaning in terms of multiplicity. Other than the fact that the proposed measure correlates with other standard measures of multiplicity, it does not have a multiplicity interpretation. The interpretation in Theorem 3.3 relies on strong assumption and it is unclear if $\epsilon'$ can make it meaningful in practice.
- Limited scope of applicability. It is unclear whether the method is only useful for LoRA or other models as well.

**Questions For Authors:**

- Why multiple rows (e.g., for the Car dataset) in Table 5 lack the bold highlight?
- What is the x axis in Figure 3 and other similar figures in the appendix?
- Do the correlation results hold outside of LoRA finetuning? E.g., other adapters or full finetuning.

**Relation To Broader Scientific Literature:**

Measuring multiplicity is an interesting problem, but it can be computationally challenging for realistic models. The paper proposes a method based on sampling a series of predictions in a neighbourhood around a point as a proxy measure that could flag high multiplicity in a way that is computationally inexpensive. However, the proposed method seems to be quite limited in scope, operational interpretation, and have limited applicability in practice due to the dependence on the arbitrary parameter $\sigma$.

**Theoretical Claims:**

I have not carefully checked the proof of the formal statement.

---

> ### Author Rebuttal · Authors · 2025-04-01
>
> We thank the reviewer for their review!
>
> Link to **PDF** with new Figures and Tables: https://drive.google.com/file/d/1zMdT0zdMrIPO9eUCHGYZ-WUlYsgbyDYu/view
>
> ---
>
> **Additional Re-trainings:** We have included an additional experiment with 100 retrainings (see Table 11 in PDF) and still see a high correlation with our Stability measure. Our core strength lies in the fact that the Stability measure, computed using just a single model, demonstrates ~90% correlation with actual multiplicity which is computationally expensive as it requires fine-tuning several models. We would like to respectfully point out that the related work [5] cited by the reviewer trains thousands of models just for logistic regression. They only re-train 50 models for neural networks on CIFAR-10 citing computational constraints. In our case, this challenge is amplified as we work with LLMs with millions of parameters. Not only is training expensive, but the cost of inference for evaluation also scales with $O(N×m)$, where $N$ is the size of the test set and $m$ is the number of retrained models, making experiments with thousands of models especially demanding in our setting. We will definitely cite this paper and include it in our discussion.
>
> ---
>
> **Choice of Sigma:** Due to character limits, we kindly refer to our response to Reviewer G1UA under **"Choice of Sigma"**.
>
> ---
>
> **Regarding Applicability Outside LoRA Method:** We focus on LoRA since it is currently the most widely adopted parameter-efficient fine-tuning method for LLMs. However, we have now added an ablation using Prompt Tuning and Prefix Tuning (see Table 12 in PDF). While our stability measure continues to correlate with multiplicity under these settings, the correlations are weaker—likely due to the limitations of these tuning methods, which are known to be less effective than LoRA in few-shot scenarios. We will include an ablation with other methods (e.g. full fine-tuning) in the revised version to study the limits of our method.
>
> ---
>
> **Adversarial weight perturbation comparison:**  We will definitely cite and elaborate the discussion on AWP in the seminal paper “Rashomon Capacity.” We have *included a new experiment, implementing our adaptation for AWP for LLMs* (as another baseline) since that paper is not tailored for LLMs. While AWP is computationally cheaper than retraining for small models [6], we found it prohibitively expensive for LLMs due to the following reason: Each AWP gradient optimization step requires full forward passes on the test set to make sure the model is in the Rashomon set, contributing to high inference costs alongside high gradient-computation costs. Our adapted AWP implementation for LLMs (16 Hrs) (see Table 11) actually took longer than training multiple models (8 Hrs) while still demonstrating weaker or similar correlation to actual multiplicity, making our stability measure (26 mins) more practical for large-scale deployment. See Fig. 9 & Table 11 in PDF link for results including a time comparison that we will also include in our final version.
>
> ---
>
> **Lack of operational meaning in terms of multiplicity:** Due to character limit, we kindly refer to our response to Reviewer G1UA under **Regarding Mean and Variability Term**.
>
> ---
>
> **Reviewers Suggestions:** Thank you for the thoughtful suggestions. We will incorporate these into the revised version of the paper. We also appreciate the naming feedback and are actively considering renaming "*stability*" to "*consistency*" to better reflect the interpretation provided in Theorem 3.3.
>
> ---
>
> **Thanks for your questions!**
>
> * Missing Bold Highlights in Table 5: We'll correct this in the revised version.
>
> * X-axis in Figure 3: The x-axis corresponds to the respective metric being plotted—Stability (blue), Prediction Probabilities (green), and Drop Pred (orange). Each curve shows how evaluated multiplicity changes as the value of that stability metric changes, with shaded regions denoting variance. To get the plot, we first group test datapoints by their stability scores (e.g., taking a small “window” of stability values at a time), then we plot the mean evaluated multiplicity (e.g., prediction range) of those datapoints, along with the standard deviation across them.
>
> * Correlation Results Beyond LoRA: Please refer to our response under **Regarding Applicability Outside LoRA Method**.
>
> ---
>
> [1] Cohen et al., 2019  Certified Adversarial Robustness via Randomized Smoothing
>
> [2] Salman et al., 2019 Provably Robust Deep Learning via Adversarially Trained Smoothed Classifiers
>
> [3] Cortes, Corinna, and Vladimir Vapnik. "Support-vector networks.
>
> [4] Ester, Martin, et al. "A density-based algorithm for discovering clusters in large spatial databases with noise.
>
> [5] Bogdan Kulynych, et.al., Arbitrary Decisions are a Hidden Cost of Differentially Private Training
>
> [6] H Hsu, et al, Rashomon capacity: A metric for predictive multiplicity in classification

---

> > ### Comment · Reviewer_3rKa · 2025-04-04
> >
> > Thanks for the response and for the additional experiments. These results with additional number of re-trained models and AWP address some of my comments.
> >
> > I still would not recommend either "stability" or "consistency", as the latter has also been used in a [different sense](https://arxiv.org/abs/2301.11562). Consider something technically descriptive along the lines of "Confidence-Variability Discrepancy" or "Local Confidence Consistency", which does not imply immediate interpretation as prediction stability.

---

> > > ### Author Response · Authors · 2025-04-09
> > >
> > > Thank you for increasing your rating! We’re glad we were able to address your comments and truly appreciate your thoughtful feedback throughout the process. We will update the name of our measure in the revised paper to improve clarity and avoid confusion.
> > >
> > > Your questions and comments have significantly strengthened our paper, and we’re very grateful for your engagement!

---

### Official Review · Reviewer_dw7t · 2025-03-17

**Overall Recommendation:** 3

**Summary:**

This paper addresses the challenge of ​fine-tuning multiplicity in tabular LLMs, where minor variations in training (e.g., seeds, hyperparameters) lead to conflicting predictions across equally performant models. The authors propose a ​stability measure, which quantifies prediction robustness by analyzing the local behavior of a single model around an input in the embedding space.

**Claims And Evidence:**

Theoretical guarantees are provided to link high stability scores to robustness across a broad class of fine-tuned models.

**Essential References Not Discussed:**

NA

**Experimental Designs Or Analyses:**

Experiments span multiple datasets (tabular and synthetic), model architectures, and fine-tuning methods. The correlation analysis with multiplicity metrics (e.g., Prediction Variance) convincingly demonstrates the measure’s practical utility.

**Methods And Evaluation Criteria:**

Extensive experiments on real-world datasets (e.g., Diabetes, German Credit) demonstrate that the proposed measure outperforms baselines like prediction confidence and dropout-based methods in capturing fine-tuning multiplicity.

**Other Comments Or Suggestions:**

NA

**Other Strengths And Weaknesses:**

Weakness:

The guarantees rely on strong assumptions (e.g., unbiased estimators, bounded gradients). Real-world fine-tuning may violate these, especially with non-LoRA methods or large distribution shifts. The paper does not empirically validate these assumptions or discuss their practical relevance.

**Questions For Authors:**

NA

**Relation To Broader Scientific Literature:**

NA

**Theoretical Claims:**

The probabilistic guarantee (Theorem 3.3) under LoRA-based fine-tuning assumptions provides a principled foundation for the stability measure. The connection between local embedding behavior and model robustness is well-motivated and theoretically justified.

---

> ### Author Rebuttal · Authors · 2025-04-01
>
> Thank you for your positive review!
>
> Link to **PDF** with new figures and tables: https://drive.google.com/file/d/1zMdT0zdMrIPO9eUCHGYZ-WUlYsgbyDYu/view
>
> ---
>
> **Regarding strong assumptions:** Our theoretical analysis draws inspiration from standard assumptions in optimization and statistical learning (e.g., bounded gradients, unbiased estimators) to derive tractable mathematical guarantees. While these assumptions may not hold exactly in practice, they are common in theoretical ML literature and serve to formalize the intuition and justification behind our stability measure. Crucially, our experiments demonstrate that the measure remains effective even when these assumptions are relaxed (e.g., under non-LoRA fine-tuning like T-Few), suggesting robustness to mild violations. The bounded gradient assumption aligns with parameter-efficient methods like LoRA, where low-rank updates naturally constrain function space deviations [1]. For large distribution shift, our guarantees would not directly apply, which we acknowledge as a limitation. However, our focus is on fine-tuning multiplicity (minor model variations) rather than large distributional changes which makes these assumptions reasonable in context.
> Our core highlights are: (i) Even for small changes to the hyperparameters, fine-tuned models do exhibit a significant amount of multiplicity; and (ii) Our measure, computed using just a single model, demonstrates ~$90$% correlation with actual multiplicity which is computationally expensive as it requires fine-tuning several models, which is our core strength. Also see (Fig. 9 and Table 11) in PDF link for a time comparison.
>
> Future work could extend this analysis to broader settings. Our method can be implemented by a practitioner without a detailed understanding of this theory. This flexibility allows our approach to be accessible to practitioners, making it a valuable tool for the community. Providing theoretical insights and motivations should be seen as an additional strength—offering deeper understanding and interpretability—rather than a drawback, and we believe such contributions should be encouraged in research.
>
> ---
>
> **Theoretical Results:** Our theoretical analysis leverages standard assumptions in optimization and statistical learning (e.g., bounded gradients, unbiased estimators) to derive tractable mathematical guarantees. While these assumptions may not hold exactly in practice, they are common in theoretical ML literature and serve to formalize the intuition and justification behind our stability measure. Crucially, our **empirical results validate the practical utility of the stability measure** across multiple datasets, model architectures, and fine-tuning recipes, demonstrating strong correlation between stability scores and evaluated multiplicity *without requiring explicit assumption or estimation of theoretical constants*.
>
> The one-sided guarantee is purposeful because our goal is to **certify when predictions are stable**. Specifically, our theoretical result ensures that if the stability score $S(x,f)$ is high, then with high probability the true prediction $F(x)$ will be at least $S(x, f) - \epsilon$. **Our proposed measure serves as a useful and informative lower bound of the model predictions $F(x)$ with a certifiably small gap**. That is, the prediction won’t fall below this threshold across well-performing fine-tuned models. This is exactly the kind of assurance we want: if the stability score is high, the prediction is robust.
>
> ---
>
> **Clarity on Figure 3:** The goal of Figure 3 is to visually show that high stability corresponds to lower multiplicity (aligning with the intuition in Thm. 3.3). We group test datapoints by their stability scores (taking a small “window” of stability values at a time along x-axis). Then, we plot the mean evaluated multiplicity (e.g., prediction range) of those datapoints, along with the standard deviation across them. Sliding this window across the entire test set shows the relationship between stability and multiplicity. This is visually neater than a scatter plot. As for Avg. Prediction Range and Avg. Prediction Variance in Table 1, they are indeed the averages over all datapoints in the test set, e.g., $\frac{1}{N} \sum_{i=1}^N  PR(x_i)$. We will clarify these details in the final version.
>
> ---
> [1]  Zeng, Y. et.al.. The expressive power of low-rank adaptation.

---

### Official Review · Reviewer_ZkiU · 2025-03-25

**Overall Recommendation:** 3

**Summary:**

This paper studies the problem of fine-tuning multiplicity in tabular LLMs, where models trained from the same pre-trained checkpoint under different conditions (e.g. random seeds) make inconsistent predictions on the same inputs. The authors propose a new measure called consistency to estimate the robustness of individual predictions without retraining multiple models. This is done by perturbing the input in the embedding space and computing the average output behavior. The paper provides a theoretical guarantee that high consistency implies stability across a broad class of similarly performing fine-tuned models. Empirical results on six tabular datasets show that the proposed score correlates well with several multiplicity metrics. The method is also compared to prediction confidence and a dropout-based baseline, and performs better in most cases.

**Claims And Evidence:**

The central claims -- that fine-tuning multiplicity is a real issue in tabular LLMs and that the proposed consistency score captures prediction robustness -- are mostly supported by solid evidence. The authors provide empirical results across multiple datasets, models, and tuning setups, and compare their method to reasonable baselines. The claim that high consistency implies stability is backed by a theoretical guarantee, though it relies on strong assumptions that are not verified or estimated in practice. The paper also shows strong correlations between the consistency measure and multiplicity metrics, but stops short of demonstrating how this score can be used in practice (e.g. for model selection or filtering). As a result, the measure remains primarily diagnostic, and the practical utility of the score is not fully established.

**Essential References Not Discussed:**

No major omissions stood out

**Experimental Designs Or Analyses:**

The experimental setup is sound and addresses the core claims of the paper. The authors evaluate their method across multiple datasets, models (T0 and FLAN-T5), and tuning strategies (T-Few and LoRA), and use standard multiplicity metrics. They also include ablation studies for key hyperparameters. However, the experiments focus only on correlation with multiplicity and do not explore downstream uses of the consistency score, such as improving model reliability or filtering unstable predictions. This limits the practical insight gained from the analysis.

**Methods And Evaluation Criteria:**

Yes

**Other Comments Or Suggestions:**

1. Figure 2 does not make sense in the context of the paper and should be motivated or better connected to the rest of the paper.
2. The authors should consider moving “Candidate Measure: Prediction confidence” from Section 3 to another subsection of Section 3.
3. The paper would benefit from a brief discussion of how the proposed measure could be used in downstream tasks

**Other Strengths And Weaknesses:**

Strengths
1. The paper introduces a simple and computationally efficient method to estimate prediction stability without retraining, which is well-motivated for tabular LLMs.
2. Theoretical analysis (Theorem 3.3) is clear and adds value, even if assumptions are satiesfied.
3. Evaluation is thorough, which covers multiple datasets, models, and fine-tuning methods.

Weaknesses:
1. The consistency measure is only evaluated in terms of correlation with multiplicity metrics -- its downstream usefulness remains unclear.
2. The model class considered is limited to variations from random seeds and does not address other practical sources of instability.
3. No methods are proposed to reduce or mitigate multiplicity; the work is diagnostic only.
4. Synthetic experiments (Figure 2) are illustrative but not well-analyzed or connected to real-world behavior.
5. Assumptions behind the theoretical guarantee are strong and cannot be verified empirically; constants in the bound are not grounded.

**Questions For Authors:**

1. Can authors provide more context about how/why each of the measures mentioned in Section 2 is useful, and why they are used in the context of this paper?
2. How is stability (Definition 3.1) different from other similar measures? I can think of some sort of adversarial robustness metric in the embedding space off the top of my head. Are there any prior works that have considered this measure? It could be in the context of other applications.

**Relation To Broader Scientific Literature:**

The paper connects well with existing work on predictive multiplicity, particularly the Rashomon effect and related metrics like arbitrariness and disagreement. It builds on recent methods for measuring multiplicity in neural networks and extends them to tabular LLMs, which are less explored. It also draws from prior work on robustness via perturbations and embedding space sampling.

**Theoretical Claims:**

The main theoretical result provides a lower bound on the prediction of a fine-tuned model in terms of the proposed consistency score. The proof appears correct and is clearly presented.

However, the bound relies on constants like alpha, t, and L that are not estimated or discussed empirically, which makes it difficult to apply in practice. The key assumption about the stochasticity of the model class is also strong and not verifiable. Plus, the guarantee is one-sided as it only certifies that predictions won't fall too far below the estimated consistency score, but provides no insight for cases where consistency is low and predictions may still be reliable. These choices are acknowledged by the authors, but they constrain how actionable the theoretical results are in practice.

---

> ### Author Rebuttal · Authors · 2025-04-01
>
> We thank the reviewer for their positive review!
>
> Link to PDF with new figures and tables: https://drive.google.com/file/d/1zMdT0zdMrIPO9eUCHGYZ-WUlYsgbyDYu/view
>
> ---
> **Regarding downstream use**: While our work focuses on quantifying the stability of predictions—to our knowledge, the first to do so for LLMs in the context of fine-tuning multiplicity—this is a necessary and foundational step for enabling trust in downstream applications. We agree that practical uses are important, and our measure would support actionable decisions: practitioners can filter out or exercise caution for the low-stability predictions, e.g., a candidate getting different decisions from similar models in a loan decision will cause reputational risk or even fairness concerns. However, they can trust predictions on data-points with high stability, which are provably robust across fine-tuned variants without actually retraining an ensemble of models. This aligns with deployment needs in high-stakes domains such as healthcare, finance, hiring, and education, where reliability often matters more than full coverage. Using stability scores to actively mitigate fine-tuning instability is an exciting direction for future work. However, our current goal is to rigorously define and validate a stability measure, both theoretically and empirically. These results lay the necessary groundwork for future methods that could leverage stability scores to guide or regularize training using more stable data points. We will add a brief discussion on these directions in the paper.
>
> ---
> **Scope of model class (W2):** Our focus on seed-induced variations aligns with prior work on predictive multiplicity [1,2,3,4], where controlled stochasticity (e.g., initialization, data shuffling) is used to isolate the impact of training randomness—a foundational and prevalent source of instability in fine-tuning. While practical deployments may encounter other sources of variation (e.g., hyperparameter changes or distribution shifts), we deliberately focus on this aspect to **highlight** that even small sources of randomness can lead to highly arbitrary predictions, as demonstrated in our experiments. Several related works on multiplicity also adopt this setting [1,2,3,4].
>
> ---
> **Motivation for multiplicity measures in Section 2 (Q1):**  We will revise the text to better clarify the role of each measure. Briefly, we include these standard multiplicity metrics—Arbitrariness, Discrepancy, Pairwise Disagreement, Prediction Variance, and Range—as they each capture different facets of prediction inconsistency across fine-tuned models. These measures serve as ground truth evaluations of fine-tuning multiplicity and allow us to benchmark the effectiveness of our proposed stability score in predicting multiplicity without retraining. Arbitrariness and Discrepancy capture label-level disagreement; Prediction Variance and Range capture the spread of softmax outputs; Pairwise Disagreement captures disagreement among all model pairs. Together, they provide a comprehensive picture of multiplicity and motivate our evaluation framework.
>
> ---
> **Novelty of the stability measure (Def 3.1):**
> While our formulation shares surface-level similarities with local robustness metrics (e.g., certified robustness uses the mean of predictions in a neighborhood), our objective and motivation are fundamentally different. Traditional robustness measures typically try to capture how resistant a model is to worst-case perturbations (e.g., adversarial examples), whereas our **stability measure tries to capture prediction consistency across a class of fine-tuned models**—a fundamentally different notion that we refer to as fine-tuning multiplicity. To the best of our knowledge, no prior work has proposed this exact formulation to quantify robustness to fine-tuning variability in LLMs or tabular settings. Unlike adversarial robustness, we sample random local perturbations in the embedding space to estimate prediction smoothness in the neighborhood, combining local confidence and variability to derive a **probabilistically guaranteed lower bound** (Thm 3.3) on prediction consistency. We will clarify this distinction in the revised draft.
>
> ---
> **Theoretical Results:** Due to character limit, we kindly refer to our response to Reviewer dw7t under **"Theoretical Results"**.
>
> ---
> [1] Gomez, et al. Algorithmic arbitrariness in content moderation
>
> [2] Hsu, H. et al.. Rashomon capacity: A metric for predictive multiplicity in classification
>
> [3] Watson-Daniels, et al. Predictive multiplicity in probabilistic classification
>
> [4] Hsu, H., et al Dropout-based rashomon set exploration for efficient predictive multiplicity estimation.

---

### Official Review · Reviewer_G1UA · 2025-03-25

**Overall Recommendation:** 3

**Summary:**

The work studies the problem of model multiplicity in LLM classification for tabular data. Model multiplicity refers to the phenomenon that multiple models of similar accuracy assign confliciting predictions to individual instances. The authors propose a measure of model multiplicity (that does not require retraining), provide some theoretical observations about it (Section 3.2), and evaluate it in six tabular datasets.

**Claims And Evidence:**

The proposed metric (called Stability) implicitly assumes that local neighborhoods in embedding space reflect nearby models in function space.
This is the key insight that allows the authors to estimate model multiplicity without retraining.

While I see no a priori reason why this should be true or false, the high Spearman correlation between Stability (which does not require retraining) and metrics such as Arbitrariness and Prediction Variance (which do require expensive retraining) is evidence that this assumption is true. That Stability has a higher Spearman correlation than other metrics that do not require retraining (e.g. Drop-Out) is a major strength and the main contribution of this work.

**Essential References Not Discussed:**

I do not see any missing related works in this paper.

**Experimental Designs Or Analyses:**

I think authors need to explain more clearly what exactly Figure 3 is plotting, namely how exactly the uncertainties in Figure 3 are computed. I may be mistaken, but from my understanding, the Stability metric is evaluated across the entire test dataset for a fixed model $f$, i.e.  $[S(x_1, f), S(x_2, f), \ldots S(x_N, f)]$. Then (using Fig 3a as an example), the predictive range is evaluated across the entire test dataset for the 40 trained models $[\text{PR}\_{\delta}(x_1), \ldots, \text{PR}_{\delta}(x_N)]$. What we see in Figure 3a is these two arrays plotted together. If my understanding is correct,where does the uncertainty come from? Is $S(x, f)$ evaluated for many different functions $f$?

I also had a question (which I repeat in the Questions section) about Table 1, namely what exactly "Avg. Prediction Range" and "Avg. Prediction Variance" means. The prediction range and prediction variance are a function of the embedding $x$ and the class prediction $c$, and it is unclear how authors averaged over these two quantities. I assume authors averaged over the dataset, using $c = \text{argmax}_{i \in [C]} f_i(x)$ for each embedding $x$, however this is not explicitly stated.

**Methods And Evaluation Criteria:**

The exact choice for the metric (Stability) confuses me in a few ways. It is worth explaining how this metric is computed, as in my opinion it is not intuitive. For a given label $x$, we assume that $f(x) = [f_1(x), \ldots, f_C(x)]$ outputs the softmax over classes, and that we predict that label $x$ belongs to class $c = \text{argmax}_{i \in [C]} f_i(x)$.

The Stability metric only looks at the predicted probability $f_{c}(x)$, and ignores all the other probabilities in the softmax. The metric is

$\frac{1}{k} \sum_{x_i \in N_{x,k}} f_c(x_i) - \frac{1}{k} \sum_{x_i \in N_{x,k}} |f_c(x_i) - f_c(x)|$. We see that the Stability metric has two contributions: a Local Averaging term and a Variability Penalization term. I do not understand why the Local Averaging term is needed. We already know that $f_c(x)$ is the largest predicted probability out of all the classes, so why do we care about its magnitude in a neighborhood around $x$? I would find just the Variability Penalization term to be a more intuitive metric, since my main concern is how much the value of $f_c(x)$ changes within a neighborhood of $x$. The magnitude does not matter because we already predicted class $c$.

My main question ( which I will repeat in the Questions Section) to the authors is why they did not consider a metric such as:

$$M_1(x) =  \frac{1}{k} \sum_{x_i \in N_{x,k}} |f_c(x_i) - f_c(x)|$$

or its square, so that this has the interpretation of a variance:

$$M_2(x) =  \frac{1}{k} \sum_{x_i \in N_{x,k}} (f_c(x_i) - f_c(x))^2.$$

What exactly does the metric $S$ capture that $M_1$ or $M_2$ miss? I also find the metric $M_2$ to be particularly pleasing since it looks very similar to the Predictive Variance metric used in the paper. Moreover, assuming the experiments do not take too long, I am curious if metrics such as $M_1$ and $M_2$ also exhibit correlation with the more expensive metrics that require retraining.

**Other Comments Or Suggestions:**

I have no further suggestions. I am leaning towards reject, as I think the paper needs some polishing and more results, particularly on a larger datasets, before it is ready for acceptance at ICML.

**Other Strengths And Weaknesses:**

One of the most intriguing plots in this paper is Figure 7 in the Appendix. This figure clearly shows that the parameter $\sigma$ (which controls how large of a neighborhood around $x$ you perturb) is the single most important parameter to tune to get the strong correlations observed in this work. Indeed, choosing $\sigma = 10^{-2}$ clearly makes the Stability metric correlated with all the other metrics that require retraining. At the same time, this correlation disappears if sigma is either too big or too small. In practice, however, one does not have the luxury of comparing against metrics that require retraining to tune $\sigma$. How do authors recommend $\sigma$ be chosen in practice? There is one offhand remark that "To guide the choice of $\sigma$, one could consider the spread of training samples", however this is not explored further. I think this is one of the most important problems that authors should address, since if $\sigma$ can be easily estimated from the training data, then this approach can and should be used in practice by all high stakes LLMs to identify problematic points in the test set. But if $\sigma$ cannot be estimated without retraining, then this approach is not useful in practice.

Assuming the above is resolved, I think the paper would be much stronger if authors used the Stability metric on a larger dataset where retraining 40 times is computationally infeasible (require days/weeks of fine tuning). Authors could use the Stability metric to identify which points on this larger dataset could be more susceptible to arbitrary decisions. Perhaps these are data-points that belong to a minority group, or are mislabeled?

**Questions For Authors:**

1. (See Methods And Evaluation Criteria for context) Why choose the Stability metric over metrics such as $M_1$ and $M_2$? Do metrics such as $M_1$ and $M_2$ also exhibit the same correlations seen with Stability?

2. (See Experimental Designs Or Analyses for context) In Table 1, what exactly is "Avg. Prediction Range" and "Avg. Prediction Variance"?

3. (See Other Strengths And Weaknesses for more context) How do authors recommend $\sigma$ be chosen in practice? Could this be applied to a larger dataset where retraining is infeasible?

**Relation To Broader Scientific Literature:**

See my answer in "Claims And Evidence".

**Theoretical Claims:**

I checked the proof of Theorem 3.3 and verified its correctness.

---

> ### Author Rebuttal · Authors · 2025-04-01
>
> We thank the reviewer for their review!
>
> Link to **PDF** with new figures and tables: https://drive.google.com/file/d/1zMdT0zdMrIPO9eUCHGYZ-WUlYsgbyDYu/view
>
> ---
> **Regarding Mean and Variability Term:** Our stability measure relies upon both local variability and mean confidence because they capture synergistic aspects of prediction robustness in classification (see motivational Fig. 8 in PDF). Model prediction confidence has long been used as a measure of reliance (roughly the distance from the decision boundary), e.g., in multi-class classification, if the output logits are [0.4,0.3,0.3] vs [0.6, 0.2, 0.2], the latter may be more robust. If the average confidence in a neighborhood is higher, it is an even stronger indicator of robustness (e.g., a prediction differing significantly from all its neighbors may be unreliable). Points near decision boundaries might exhibit higher arbitrariness and pairwise disagreement than those in high-confidence neighborhoods (for classification). However, the mean confidence alone is not sufficient if the point lies in a highly fluctuating region, e.g., two neighborhoods with identical variability ≈$0.05$ but markedly different mean confidences $0.8$ vs $0.33$: the former is still more likely to be in the same class. While just the variability term quantifies local fluctuations, the mean confidence provides crucial information about base neighborhood prediction confidence. We now include an additional experiment, with the variability term $M_1(x)$ and $M_2(x)$, showing a weaker correlation than our measure (Table. 10 in PDF).
>
> ---
> **Choice of Sigma:** While our ablation studies showed that $\sigma=10^{-2}$ achieved optimal results, we found that nearby values $10^{-3}$ to $10^{-1}$ still maintained strong correlations with multiplicity metrics (Fig. 7), demonstrating tolerance to the exact choice of $\sigma$. Drawing inspiration from related domains such as kernel methods [3,4] or certified robustness [1,2] which involve a similar neighborhood hyperparameter, we suggest the following **practical, data-driven method to set $\sigma$ without retraining**:
>
> * Compute Pairwise Distances: For all training samples, calculate the median distance $d_{med}$ between each point and its k-nearest neighbors ($k=5$) in the embedding space.
>
> * Set $\sigma$ as a Fraction of $d_{med}$: Choose $\sigma = 0.1 d_{med}$. This captures the natural scale of the data while ensuring perturbations stay within the local neighborhood.
>
> We tested this on Diabetes and Adult datasets. The computed $d_{med}$ was ~0.1, leading to $ \sigma = 0.01$, which matched the optimal ablation value (Fig. 7). This method generalizes across datasets without retraining.  $ d_{med} $ reflects the inherent data density—perturbations smaller than this preserve local structure, while larger values risk overshooting. Several ML methods use similar dataset statistics e.g., median(pairwise distances) approaches to set parameters (e.g., DBSCAN [4], Kernel Methods / RBF Kernels [3], etc). Importantly, this suggested method for choosing $\sigma$—based on distance to nearest neighbors in the training set—requires no model retraining and hence scales seamlessly to larger datasets.  We will include this procedure in the revised manuscript.
>
> Related fields like certified robustness rely on similar choices of hyperparameter $\sigma$ between $10^{-2}$ to $10^{-1}$, often guided by dataset scale or empirical tuning [1,2].  We note that our baseline methods such as Drop-out [Hsu et al 2024] also rely on selection of hyperparameters such as the drop-out rate (heuristic or search-based).
>
> ---
> **Larger Dataset:**  We have now included runtime experiments on the Adult dataset (see Fig. 9 and Table 11 in PDF), a dataset which contains over 40k samples and represents a setting where retraining multiple models is computationally expensive. Specifically, evaluating multiplicity by retraining 100 fine-tuned models on this dataset took over 8 hours, and another baseline, AWP, required 16 hours. In contrast, our method (stability) just took 26 mins, since it does not require retraining. Despite a drastic reduction in runtime, our method achieved higher correlation with multiplicity metrics compared to baselines methods. See bar chart in Fig. 9 and detailed training and evaluation runtimes in Table 11 in PDF. Our core strength lies in the fact that our measure, computed using just a single model, demonstrates high correlation with actual multiplicity which is computationally expensive as it requires fine-tuning several models.
>
> ---
> **On Figure 3:** We kindly refer to our response to Reviewer dw7t under **"Clarity on Figure 3"**.
>
> ---
> 1 Cohen et al, Certified Adversarial Robustness via Randomized Smoothing
>
> 2 Salman et al, Provably Robust Deep Learning via Adversarially Trained Smoothed Classifiers
>
> 3 Cortes et al, Support-vector networks
>
> 4 Ester et al, A density-based algorithm for discovering clusters in large spatial databases with noise

---

> > ### Comment · Reviewer_G1UA · 2025-04-02
> >
> > tldr: Thank you for responding to all my questions. I have bumped my score up to weak accept. In my opinion, the main thing holding this paper back is the lack of results with larger tabular datasets. I elaborate more on specifics below.
> >
> > 1) __Regarding Mean and Variability Term:__ thank you for answering my questions. I found the experimental results very convincing. Table 10 clearly shows that the Stability metric is capturing something that my suggested metrics $M_1$ and $M_2$ are not, and that "something" is the mean confidence.
> >
> > 2) __Choice of Sigma:__ Thank you for proposing a way to estimate $\sigma$ and rerunning experiments with this new approach. My only nitpick boils down to semantics. Authors stated that the method "requires no model retraining and hence scales seamlessly to larger datasets". I would slightly rephrase this as "scales seamlessly to larger datasets relative to model retraining". I nitpick this because the proposed approach requires computing $k = 5$ nearest neighbors, which, as far as I am aware, takes $O(n^2)$ time, especially in high dimensions. I would not call this approach "scalable" in a vacuum. Of course, this is a nitpick because kNN will run much faster than model retraining in practice.
> >
> > 3) __Larger Dataset:__ Thank you for including runtime comparisons for Adult, however this is not what I had in mind. I was curious about running the stability metric on datasets that would take ~days/weeks to retrain 40 times. Note that I am only interested in seeing the stability metric on larger datasets, not for authors to retrain 40 times on these datasets.
> > The datasets I had in mind where, for example, the datasets with >100_000 observations from [TableShift](https://github.com/mlfoundations/tableshift/tree/main).
> >
> > 4) __On Figure 3:__ Thank you for the clarification!

---

> > > ### Author Response · Authors · 2025-04-09
> > >
> > > We thank you for increasing your rating and engaging throughout this process! Thank you also for clarifying your expectations regarding large-scale evaluations. **We have now included experiments on the Hospital-Readmission Dataset from [TableShift](https://github.com/mlfoundations/tableshift/tree/main) ([UCL](https://archive.ics.uci.edu/dataset/296/diabetes+130-us+hospitals+for+years+1999-2008)), which consists of 101,766 data points and 47 features.**
> > >
> > > We first trained a single model, which took 210.2 minutes (~3.5 hours). Note that *training 40 such models would take approximately 5 days*, making retraining-based multiplicity estimation prohibitively expensive. In contrast, our stability measure requires only a single model and is thus ideal for such large-scale deployments. We use a 80:20 train test split ratio.
> > >
> > >
> > > In the absence of multiple retrained models, we evaluate our method by analyzing the average stability, prediction confidence, and dropout-based confidence for correctly and incorrectly classified samples and their respective runtimes.
> > > The results are summarized below:
> > >
> > >  ---
> > >
> > > | Method                | Correctly Classified |                      | Incorrectly Classified |         |
> > > |-----------------------|----------------------|----------------------|------------------------|----------------------|
> > > |                            | Mean                 | Std                  | Mean                   | Std                  |
> > > | Stability             | 0.8710               | 0.1465               | 0.5729                 | 0.1458               |
> > > | Prediction confidence| 0.8994               | 0.2160               | 0.7965                 | 0.2256    |
> > > | Dropout               | 0.8190               | 0.2832               | 0.7217                 | 0.1929   |
> > >
> > > **Table A**: *Mean and std of stability, prediction confidence, and dropout scores for correct vs. incorrect predictions.*
> > >
> > > ---
> > >
> > > We observe that correctly classified points consistently exhibit higher stability and confidence across all methods. Notably, our stability score shows a larger gap between correct and incorrect predictions compared to the dropout-based uncertainty, suggesting it is better at discriminating against unreliable predictions. We include our runtime comparisons in Table C.
> > >
> > > ---
> > >
> > > | Confidence | Stability | Description | % Test set|
> > > |----------------|----------------|-----------------------------------|----------|
> > > | High (≥ 0.75) | High (≥ 0.75) | Confident & Stable (good) |41% |
> > > | High (≥ 0.75) | Low (< 0.75) | Confident but Unstable ❗ |20% |
> > > | Low (< 0.75) | High (≥ 0.75) | Unconfident but Stable |22% |
> > > | Low (< 0.75) | Low (< 0.75) | Unconfident & Unstable (bad) |17% |
> > >
> > > **Table B**: *Test set breakdown by confidence and stability thresholds (≥ 0.75).*
> > >
> > > ---
> > >
> > > We can use our measure to analyze data points that are both confident and stable, or identify those that appear confident but are actually unstable.  In Table B, we grouped predictions based on their confidence and stability scores (for a given threshold). Observe that while 41% of the predictions were both confident and stable (ideal), a notable 20% of predictions were **confident yet unstable**—indicating that high confidence alone is not a reliable indicator of robustness.
> > >
> > > Our measure enables such fine-grained analysis, offering practitioners a tool to assess not just how confident a model is, but how consistent that confidence is across plausible fine-tuned variants. Beyond evaluating individual predictions, our measure can be used to *preemptively analyze whether certain groups are more prone to multiplicity*, helping uncover potential biases. It can also guide *data selection by filtering out low-stability points*, which improves overall model accuracy and reliability—especially valuable in high-stakes settings.
> > >
> > > To further push the limits of our method, we are willing to include results on an even larger TableShift dataset e.g, *Hypertension*, which contains 846,761 samples—in the revised manuscript. This would further demonstrate the practicality of our method in real-world, large-scale tabular learning scenarios, where retraining is infeasible and robust evaluation is essential.
> > >
> > > We sincerely thank the reviewer for their thoughtful engagement and constructive feedback throughout the review process. Your questions, comments, and suggestions have significantly helped improve the depth, and rigor of our work. We are committed to incorporating all necessary changes and believe the revised version will be much stronger as a result.
> > >
> > > ---
> > >
> > > | Metric                 | Runtime |
> > > |------------------------|----------|
> > > | Training model time    |    210.2 mins      |
> > > | Stability      |    8.8 hrs      |
> > > | Prediction confidence  | 29 mins         |
> > > | Dropout     |       19.8 hrs   |
> > >
> > > **Table C**: *Runtime comparison for methods on 20k test samples.*
> > >
> > > ---

---

### Decision · Program_Chairs · 2025-05-01

**Decision:**

Accept (poster)

**Comment:**

The paper addresses the timely problem of LLM multiplicity for tabular data. All reviewers were ultimately content with the paper and the rebuttal responses provided by the authors. including the soundness of the proposed approach and strength of the experimental results.  There are several additional suggestions that would be good to incorporate in the final version.